# Influence of Electrostatic Force Nonlinearity on the Sensitivity Performance of a Tapered Beam Micro-Gyroscope Based on Frequency Modulation

**DOI:** 10.3390/mi14010211

**Published:** 2023-01-14

**Authors:** Kunpeng Zhang, Jianwei Xie, Shuying Hao, Qichang Zhang, Jingjing Feng

**Affiliations:** 1Tianjin Key Laboratory for Advanced Mechatronic System Design and Intelligent Control, School of Mechanical Engineering, Tianjin University of Technology, Tianjin 300384, China; 2National Demonstration Center for Experimental Mechanical and Electrical Engineering Education, Tianjin University of Technology, Tianjin 300384, China; 3Tianjin Key Laboratory of Nonlinear Dynamics and Control, School of Mechanical Engineering, Tianjin University, Tianjin 300072, China

**Keywords:** frequency modulation, tapered beam micro-gyroscope, electrostatic force nonlinearity, invariant manifold method, sensitivity

## Abstract

Electrostatic force nonlinearity is widely present in MEMS systems, which could impact the system sensitivity performance. The Frequency modulation (FM) method is proposed as an ideal solution to solve the problem of environmental fluctuation stability. The effect of electrostatic force nonlinearity on the sensitivity performance of a class of FM micro-gyroscope is investigated. The micro-gyroscope consists of a tapered cantilever beam with a tip mass attached to the end. Considering the case of unequal width and thickness, the motion equations of the system are derived by applying Hamilton’s principle. The differential quadrature method (DQM) was used to analyze the micro-gyroscope’s static deflection, pull-in voltage, and natural frequency characteristics. We observed that from the onset of rotation, the natural frequencies of the drive and sense modes gradually split into a pair of natural frequencies that were far from each other. The FM method directly measures the angular velocity by tracking the frequency of the drive and sense modes. Then, based on the linear system, the reduced-order model was used to analyze the influence of the shape factor and DC voltage on the sensitivity performance. Most importantly, the nonlinear frequency of system was obtained using the invariant manifold method (IMM). The influence of electrostatic force nonlinearity on the performance of the FM micro-gyroscope was investigated. The results show that the different shape factors of width and thickness, as well as the different DC voltages along the drive and sense directions, break the symmetry of the micro-gyroscope and reduce the sensitivity of the system. The sensitivity has a non-linear trend with the rotation speed. The DC voltage is proportional to the electrostatic force nonlinearity coefficient. As the DC voltage gradually increases, the nonlinearity is enhanced, resulting in a significant decrease in the sensitivity of the micro-gyroscope. It is found that the negative shape factor (width and thickness gradually increase along the beam) can effectively restrain the influence of electrostatic force nonlinearity, and a larger dynamic detection range can be obtained.

## 1. Introduction

Micro-electro-mechanical systems (MEMS) are miniature devices that combine integrated circuits and micro-mechanical components using micro/nanofabrication techniques [1,2,3,4]. A micro-gyroscope based on MEMS technology is a kind of inertial sensor for measuring angular velocity or angular displacement, which has the advantages of low cost, small size, high durability, and mass production. It is widely used in aerospace, intelligent robotics, active safety, and consumer electronics [5,6,7,8]. Therefore, it is of great significance to study the problem of MEMS gyroscopes.

The detection method of vibrating micro-gyroscope is divided into amplitude modulation (AM) and frequency modulation. The AM micro-gyroscope [9,10] is a kind of sensor for measuring angular velocity indirectly through the Coriolis effect [11]. When the angular velocity is present, the sense mode generates a secondary vibration due to the Coriolis force. This vibration causes a change in the capacitance of the sense capacitor, and the angular velocity can be obtained by measuring the amount of capacitance change. The FM method [12] is used to perform frequency-based input angular velocity measurements by tracking the frequency split between the drive and sense modes. The angular velocity can be obtained directly from the vibration frequency [13]. The control system can be easily digitized without consuming A/D conversion resources. Moreover, FM is an innovative operating principle for micro-gyroscopes and an ideal solution to address the stability of environmental fluctuations [14].

Most micro-gyroscopes are based on the AM mode for angular velocity measurements, where the quality factor and mode-matching are the key factors affecting the sensitivity of the AM micro-gyroscope [15]. In order to improve the sensitivity performance, the conventional micro-gyroscopes often use vacuum packaging to increase the quality factor and mode-matching control by electrostatic tuning. However, the bandwidth performance is usually poor. Although micro-gyroscopes can use a complex structural design and compensation technology to achieve the navigation level requirements [16], the production costs will inevitably increase. Rather, the FM micro-gyroscope can avoid the drawbacks of the tradeoff between bandwidth and sensitivity of the AM micro-gyroscope. The FM has a good inhibitory effect on the drift mechanism of the resonant frequency and quality factor. A good detection performance can be maintained in harsh environments (vibration, shock and electromagnetic interference). Moreover, the FM micro-gyroscope has the advantages of higher scale factor stability, symmetry, sensitivity, linearity, etc. It has attracted increasing attention and will become a better choice for high-performance commercial gyroscopes in the future [17,18,19].

The FM micro-gyroscope has been in development for over a decade. As early as 2006, Moussa et al. [20] investigated a class of cantilever beam micro-gyroscopes with two vibration modes of in-plane and out-of-plane and proposed the concept of the FM gyroscope, widely used today for the first time. Zotov et al. [21,22,23,24,25] proposed a mechanical FM micro-gyroscope based on angular velocity variation, which solves the tradeoff between the gain-bandwidth and dynamic range of the traditional AM gyroscope. Their micro-gyroscope is characterized by vacuum packaging and a fully symmetrical four-proof mass structure, which improves the signal-to-noise ratio and has unlimited bandwidth. However, their research focused on the control system and manufacturing process without considering the effect of frequency mismatch on the FM detection performance. Kline et al. [26] studied a micro-gyroscope based on an orthogonal FM mode of operation and considered that the system’s sensitivity depends mainly on the angular gain factor; that is, a parameter related to the geometry of the micro-gyroscope. For the common micro-gyroscope structures, the value of the angular gain factor is between 0 and 1 [27,28]. However, the above-mentioned study did not analyze the pattern of the effect of the shape of the micro-gyroscope structure on the angular gain factor. In this study, the effect of the shape factor of the micro-gyroscope on the angular gain factor is analyzed. Effa [29] investigated a new type of FM cantilever micro-gyroscope and compared it with a crab leg gyroscope. The results showed that the cantilever micro-gyroscope has a simple control system, good bias stability, and bias repeatability. Based on a class of cantilever micro-gyroscopes with a proof mass attached at the end, Nayfeh et al. [30] proposed a method to measure the angular velocity of the base by measuring the difference between the natural frequency of the sense and the drive directions. Their results showed that the frequency-domain method is not limited to the square section beam, but is suitable for any prismatic cantilever structure with two symmetrical axes. Subsequently, Ghommem et al. [31] studied the performance of a nanocrystalline silicon cantilever micro-gyroscope based on the frequency mode. By analyzing the effects of the nanocrystalline silicon particle size and DC voltage on the calibration curve, the design scheme of the micro-gyroscope was given. It was shown that the size of the nanocrystal silicon and the fringing field had fewer effects on the calibration curve. The differential DC voltage in the drive and sense directions would destroy the symmetry of the micro-gyroscope and thus reduce the sensitivity of the system. However, the FM performance of the cantilever beam micro-gyroscope with a rectangular cross-section is not described in detail, and the effect of electrostatic force nonlinearity is not considered. Leoncini et al. [32] reported a complete FM gyroscope system with a digital output for future low-power, high-stability applications. The gyroscope exhibited good scale range and stability without applied calibration and compensation.

The cantilever structure greatly reduces the interference signals caused by vibration shocks and encapsulated stresses [29]. The tapered beams with a variable-section have good weight and strength distribution properties and are widely used in aerospace, mechanical, and construction fields. Moreover, the design and machining technology of the variable cross-section beam is becoming increasingly mature, which provides the possibility for its application in the field of micromachinery [33,34].

It is worth noting that there are some unavoidable nonlinear factors in the micro-gyroscope, such as electrostatic force nonlinearity, geometric nonlinearity, and the fringing field. These nonlinearities may produce measurement errors or even lose efficacy for the cantilever beam micro-gyroscope based on a linear dynamics design. Therefore, it is important to consider the influences of nonlinearity on the performance of the micro-gyroscope. Ghommem et al. [35] considered the electrostatic force nonlinearity in the mathematical model of a micro-gyroscope. Mojahedi et al. [36] developed the governing equations of a micro/nano-gyroscope by taking into account geometric nonlinearity, electrostatic force nonlinearity, and intermolecular forces. The influence of nonlinearity on the static deflection, pull-in instability, natural frequency, and dynamic deflection of the micro-gyroscope system was investigated. Ghayesh et al. [37] considered the size-dependent effects and applied a modified coupled stress theory to establish the governing equations of motion for a cantilever beam micro-gyroscope. Li et al. [38,39] analyzed the free vibration response of a piezoelectric cantilever beam based on the IMM [40,41,42]. The influence of the gyroscopic effect caused by the rotational angular velocity on the frequency of the system was studied. The nonlinear frequency and nonlinear mode of the system are calculated by the method of multi-scale, and the nonlinear mode motion of the piezoelectric cantilever beam at different rotational speeds is studied. However, their study did not apply the electrostatic force nonlinearity to the FM micro-gyroscope.

The establishment of a more accurate mathematical model is the basis for analyzing and improving the performance of the micro-gyroscope. Most of the above research focused on the influences of nonlinear factors on the dynamic response of an AM micro-gyroscope, while the influence analysis of nonlinearity on the FM micro-gyroscope is still lacking. It is well known that the nonlinear softening (or hardening) characteristics cause the natural frequency to change, which induces the measurement errors of the FM micro-gyroscope based on linear frequency difference detection. Therefore, the electrostatic force nonlinearity is considered in this work. We find that the electrostatic force nonlinearity shifts the natural frequency of the drive and sense directions. The stronger the nonlinearity, the greater the degree of the natural frequency shift, and the nonlinearity will significantly reduce the sensitivity of the FM micro-gyroscope. In addition, the FM micro-gyroscope not only has a greater dynamical range, but is also least affected by nonlinearity when the shape factor of the tapered beam is negative. The main contribution of this paper is to investigate the influences of electrostatic force nonlinearity on the dynamic range and sensitivity of the FM micro-gyroscope and provide theoretical guidance for the design of the FM micro-gyroscope.

Symbol Table.*α*_4_, *α*_5_*α*_4_ is the shape factor of the thickness, *α*_5_ is the shape factor of the width
*L*
Length of beam*a*(*x*)Thickness of beam at length *x**b*(*x*)Width of beam at length *x**v*(*x*, *t*)Bending displacement in the *y*-direction at any section relative to the base frame*w*(*x*, *t*)Bending displacement in the *z*-direction at any section relative to the base frame
*M*
Mass of the tip mass block*m*(*x*)Cross-sectional mass at beam length *x*
*J_ξξ_*
Moment of inertia around the *ξ*-axis
*J_ηη_*
Moment of inertia around the *η*-axis
*J_ζζ_*
Moment of inertia around the *ζ*-axis
*ρ*
Material density of beam
*E*
Young’s modulus of beam*I_η__η_* (*x*)At length *x*, the cross-section second moment of area about the *η*-axis*I_ζζ_* (*x*)At length *x*, the cross-section second moment of area about the *ζ*-axis
*b_M_*
Width of tip mass block
*h_M_*
Thickness of tip mass block
*d_w_*
Initial gap of capacitor in the drive direction
*d_v_*
Initial gap of capacitor in the sense direction
*ε*
_0_
Dielectric constant
*A_w_*
Capacitor area in the drive direction
*A_v_*
Capacitor area in the sense direction
*β_F_*
Fringing field parameters [43]
*ω*
_0_
External excitation frequency
*V_w_*
Voltage in the drive direction
*V_v_*
Voltage in the sense direction
*V_AC_*
The amplitude of the AC voltage applied in the drive directionΩInput angular velocity along *x*-axis. The physical quantity is measured according to the frequency of the split.*V_DC_*_1_, *V_DC_*_2_Bias DC voltage amplitude along the drive and sense directions, provided by the capacitor below the tip mass. It affects the static pull-in voltage, natural frequency, and nonlinear coefficient of the system.ΓMotion amplitude of the system*w_d_*(*x*, *t*)Dynamic displacement in the *z*-direction at any section relative to the base frame*v_d_*(*x*, *t*)Dynamic displacement in the *y*-direction at any section relative to the base frame
*w_s_*
Static displacement in the *z*-direction relative to the base frame
*v_s_*
Static displacement in the *y*-direction relative to the base frame

κ

Angular gain factor, a physical quantity related to the geometry of the micro-gyroscope

The rest of the work in this paper is organized as follows. First, the governing equations of the system are established using Hamilton’s principle. In Section 3, the static deflection and pull-in voltage of the micro-gyroscope is analyzed based on the DQM. Then, in Section 4, the system’s natural frequency and modal shape are investigated and the effect of the angular velocity on the natural frequencies of the drive and sense modes is simulated using the DQM. In Section 5, the effects of the bias DC voltage and shape factors on the performance of the FM micro-gyroscope are discussed for a linear system. Finally, in Section 6, considering the electrostatic force nonlinearity, the IMM is used to decouple the reduced-order mode. The influence of electrostatic nonlinearity on the sensitivity of FM micro-gyroscope is analyzed.

## 2. Dynamical Model of a Tapered Beam Micro-Gyroscope

The micro-gyroscope consists of an electrostatically driven tapered cantilever beam with a tip mass attached to its free end, as shown in Figure 1. The width and thickness of the beam vary linearly along the axial direction of the beam. The thickness and width of the tapered beam at the fixed end are *a*_0_ and *b*_0_, and at the free end are *a*_1_ and *b*_1_, respectively. *α*_4_ and *α*_5_ are the shape factors of thickness and width, which describe the taper and direction of the beam, as defined in Figure 1. When *α*_4_ (*α*_5_) < 0, the thickness (width) of the beam increases linearly along the axial direction of the beam; when *α*_4_ = *α*_5_ = 0, the beam is prismatic; when 0 < *α*_4_ (*α*_5_) < 1, the thickness (width) of the beam decreases linearly along the axial direction of the beam.

In this paper, the length of the beam is 400 μm and the width *a*_0_ and thickness *b*_0_ are 2.8 μm, with a large span. The shear and torsional deformations are not considered, only the bending of the beam is considered. Therefore, it conforms to the assumption of the Euler-Bernoulli beam.

As shown in Figure 1, the *y* direction is the sense mode and the *z* direction is the drive mode of the micro-gyroscope. The static deflection of the beam can be caused by applying a DC voltage to the capacitors in the drive and sense directions. Meanwhile, an AC voltage excitation is applied in the *z* direction, which causes the beam to vibrate along the drive direction. As the angular velocity Ω*_x_* increases, the gyroscopic effect causes a gradual splitting of the natural frequencies of the drive and sense modes. Thus, the angular velocity Ω*_x_* can be obtained by measuring the difference between the splitting frequencies.

The translational kinetic energy *K*_1_, the rotational kinetic energy *K*_2_ of the tapered beam, and the translational kinetic energy *K*_3_ of the tip mass can be expressed as [44,45]:(1)$$\begin{array}{ll} K_{1}+K_{2}+K_{3}= & \frac{1}{2}\int_{0}^{L}\left(m(x\right){\left(\dot{v}-\Omega_{x}w\right)}^{2}+m(x){\left(\dot{w}+\Omega_{x}v\right)}^{2})dx\\ & +\frac{1}{2}\int_{0}^{L}\left(J_{\xi \xi }(x\right){\left(\Omega_{x}+w^{\prime }{\dot{v}}^{\prime }\right)}^{2}+J_{\eta \eta }(x){\left(\Omega_{x}\left(-v^{\prime }\right)-{\dot{w}}^{\prime }\right)}^{2}\\ & +J_{\zeta \zeta }(x){\left(\Omega_{x}\left(-w^{\prime }\right)+{\dot{v}}^{\prime }\right)}^{2}+J_{\xi \xi }(x)\Omega_{x}{}^{2})dx\\ & +\frac{1}{2}M\Omega_{x}{}^{2}(v_{L}{}^{2}+w_{L}{}^{2})+M\Omega_{x}(v_{L}{\dot{w}}_{L}-w_{L}{\dot{v}}_{L})+\frac{1}{2}M({\dot{v}}_{L}{}^{2}+{\dot{w}}_{L}{}^{2}) \end{array}$$

The coordinate system (*ξ*, *η*, *ζ*) is obtained by rotating the coordinate system (*x*, *y*, *z*) three times, as shown in Figure 2. For a more specific coordinate transformation process, please refer to the literature [7]. The expressions for the parameters in Equation (1) are as follows:(2)$$\begin{array}{l} m(x)=\rho a(x)b(x),\;a(x)=a_{0}(1-\alpha_{4}\frac{x}{L}),\;b(x)=b_{0}(1-\alpha_{5}\frac{x}{L}),\;m_{0}=\rho a_{0}b_{0}\\ J_{\xi \xi }(x)=\frac{m(x)}{12}(a{\left(x\right)}^{2}+b{\left(x\right)}^{2}),\;J_{\eta \eta }(x)=\frac{m(x)}{12}b{\left(x\right)}^{2},\;J_{\zeta \zeta }(x)=\frac{m(x)}{12}a{\left(x\right)}^{2} \end{array}$$

The elastic bending potential energy *U*_1_ of the tapered beam can be expressed as:(3)$$\begin{array}{c} U_{1}=\frac{1}{2}\int_{0}^{L}EI_{\eta \eta }(x){\left(w^{\prime \prime }\right)}^{2}dx+\frac{1}{2}\int_{0}^{L}EI_{\zeta \zeta }(x){\left(v^{\prime \prime }\right)}^{2}dx,\\ I_{\eta \eta }(x)=\frac{a_{0}b_{0}{}^{3}}{12}(1-\alpha_{4}\frac{x}{L}){\left(1-\alpha_{5}\frac{x}{L}\right)}^{3},I_{\zeta \zeta }(x)=\frac{a_{0}{}^{3}b_{0}}{12}{\left(1-\alpha_{4}\frac{x}{L}\right)}^{3}(1-\alpha_{5}\frac{x}{L}) \end{array}$$

Considering the fringing field effect, the electrostatic potential energy *U*_2_ can be expressed as:(4)$$U_{2}=\frac{1}{2}\varepsilon_{0}A_{v}\frac{V_{v}{}^{2}}{\left(d_{v}-v_{L}\right)}\left(1+\beta_{F}\frac{d_{v}-v_{L}}{h_{M}}\right)+\frac{1}{2}\varepsilon_{0}A_{w}\frac{V_{w}{}^{2}}{\left(d_{w}-w_{L}\right)}\left(1+\beta_{F}\frac{d_{w}-w_{L}}{b_{M}}\right)$$
where *V_w_* = (*V_AC_* cos(*ω*_0_*t*) + *V_DC_*_1_)^2^, *V_v_* = *V_DC_*_2_^2^.

The virtual work generated by the damping can be expressed as:(5)$$\delta W_{D}=-c\int_{0}^{L}(\dot{w}\delta w+\dot{v}\delta v)dx$$
where *c* is the damping coefficient.

The total kinetic energy, total potential energy, and virtual work are substituted into the Hamilton principle [46], that is:(6)$$\delta \int_{t_{1}}^{t_{2}}\left((K_{1}+K_{2}+K_{3})-\left(U_{1}+U_{2}\right)+W_{D}\right)dt=0$$

Then, the governing equations of the micro-gyroscope can be expressed as:(7)$$\begin{array}{c} -{\left(EI_{\zeta \zeta }(x)v^{\prime \prime }\right)}^{\prime \prime }+m(x)\Omega_{x}{}^{2}v+2m(x)\Omega_{x}\dot{w}-m(x)\ddot{v}+{\left(J_{\zeta \zeta }(x){\ddot{v}}^{\prime }\right)}^{\prime }-{\left(J_{\eta \eta }(x)\Omega_{x}{}^{2}v^{\prime }\right)}^{\prime }\\ -{\left(J_{\zeta \zeta }(x)\Omega_{x}{\dot{w}}^{\prime }\right)}^{\prime }-{\left(J_{\eta \eta }(x)\Omega_{x}{\dot{w}}^{\prime }\right)}^{\prime }+{\left(J_{\xi \xi }(x)\Omega_{x}{\dot{w}}^{\prime }\right)}^{\prime }-c\dot{v}=0\\ -{\left(EI_{\eta \eta }(x)w^{\prime \prime }\right)}^{\prime \prime }+m(x)\Omega_{x}{}^{2}w-2m(x)\Omega_{x}\dot{v}-m(x)\ddot{w}+{\left(J_{\eta \eta }(x){\ddot{w}}^{\prime }\right)}^{\prime }-{\left(J_{\zeta \zeta }(x)\Omega_{x}{}^{2}w^{\prime }\right)}^{\prime }\\ +{\left(J_{\zeta \zeta }(x)\Omega_{x}{\dot{v}}^{\prime }\right)}^{\prime }+{\left(J_{\eta \eta }(x)\Omega_{x}{\dot{v}}^{\prime }\right)}^{\prime }-{\left(J_{\xi \xi }(x)\Omega_{x}{\dot{v}}^{\prime }\right)}^{\prime }-c\dot{w}=0 \end{array}$$
where the boundary condition at *x* = 0 are:(8)v(0)=0, w(0)=0
(9)$$v^{\prime }(0)=0,\;w^{\prime }(0)=0$$
and at *x* = *L* are:(10)$$EI_{\eta \eta }(x)w^{\prime \prime }=0,\;EI_{\zeta \zeta }(x)v^{\prime \prime }=0$$
(11)$$\begin{array}{c} {\left(EI_{\zeta \zeta }(x)v^{\prime \prime }\right)}^{\prime }-J_{\zeta \zeta }(x){\ddot{v}}^{\prime }+J_{\eta \eta }(x)\Omega_{x}{}^{2}v^{\prime }+J_{\eta \eta }(x)\Omega_{x}{\dot{w}}^{\prime }+J_{\zeta \zeta }(x)\Omega_{x}{\dot{w}}^{\prime }\\ -J_{\xi \xi }(x)\Omega_{x}{\dot{w}}^{\prime }+M\Omega_{x}{}^{2}v+M\Omega_{x}\dot{w}+M\Omega_{x}\dot{w}-M\ddot{v}=0\\ {\left(EI_{\eta \eta }(x)w^{\prime \prime }\right)}^{\prime }-J_{\eta \eta }(x){\ddot{w}}^{\prime }+J_{\zeta \zeta }(x)\Omega_{x}{}^{2}w^{\prime }-J_{\eta \eta }(x)\Omega_{x}{\dot{v}}^{\prime }-J_{\zeta \zeta }(x)\Omega_{x}{\dot{v}}^{\prime }\\ +J_{\xi \xi }(x)\Omega_{x}{\dot{v}}^{\prime }+M\Omega_{x}{}^{2}w-M\Omega_{x}\dot{v}-M\Omega_{x}\dot{v}-M\ddot{w}=0 \end{array}$$
where Equations (8) and (9) represent the deflection and the rotation angle of the tapered beam at the fixed end, respectively. Equations (10) and (11) represent the flexural torque and shear force of the beam at the free end, respectively.

In Equations (7)–(11), the prime and dot denote the derivatives with respect to *x* and *t*, respectively.

The following dimensionless parameters are introduced:(12)$$\begin{array}{c} x^{*}=\frac{x}{L},v^{*}=\frac{v}{d_{v}},w^{*}=\frac{w}{d_{w}},t^{*}=t\tau ,\Omega^{*}=\frac{\Omega_{x}}{\tau },\alpha_{w}{}^{*}=\frac{\varepsilon_{0}A_{w}L^{3}}{2EI_{01}d_{w}{}^{3}},\alpha_{v}{}^{*}=\frac{\varepsilon_{0}A_{v}L^{3}}{2EI_{02}d_{v}{}^{3}},h_{M}{}^{*}=\frac{h_{M}}{d_{v}},b_{M}{}^{*}=\frac{b_{M}}{d_{w}}\\ M_{r}=\frac{M}{m_{0}L},\tau =\sqrt{\frac{EI_{01}}{m_{0}L^{4}}},d=\frac{d_{v}}{d_{w}},m(x^{*})=\frac{m(x)}{m_{0}},I_{\eta \eta }(x^{*})=\frac{I_{\eta \eta }(x)}{I_{01}},I_{\zeta \zeta }(x^{*})=\frac{I_{\zeta \zeta }(x)}{I_{02}},J_{\eta \eta }(x^{*})=\frac{J_{\eta \eta }(x)}{J_{0}}\\ J_{\zeta \zeta }(x^{*})=\frac{J_{\zeta \zeta }(x)}{J_{0}},c^{*}=\frac{c}{m_{0}\tau },I_{01}=\frac{a_{0}b_{0}{}^{3}}{12},I_{02}=\frac{a_{0}{}^{3}b_{0}}{12},I_{01}=I_{02},A_{0}=a_{0}b_{0},J_{0}=\rho A_{0}L^{2} \end{array}$$

Ignoring the asterisk for simplicity, the dimensionless governing equations can be written as:(13)$$\begin{array}{c} I_{\eta \eta }(x)w^{\left(4\right)}+2I_{\eta \eta }{}^{\prime }(x)w^{\prime \prime \prime }+I_{\eta \eta }{}^{\prime \prime }(x)w^{\prime \prime }+c\dot{w}-m(x)\Omega^{2}w+2m(x)\Omega \dot{v}+m(x)\ddot{w}\\ -J_{\eta \eta }(x){\ddot{w}}^{\prime \prime }-J_{\eta \eta }{}^{\prime }(x){\ddot{w}}^{\prime }+J_{\zeta \zeta }(x)\Omega^{2}w^{\prime \prime }+J_{\zeta \zeta }{}^{\prime }(x)\Omega^{2}w^{\prime }=0\\ I_{\zeta \zeta }(x)v^{\left(4\right)}+2I_{\zeta \zeta }{}^{\prime }(x)v^{\prime \prime \prime }+I_{\zeta \zeta }{}^{\prime \prime }(x)v^{\prime \prime }+c\dot{v}-m(x)\Omega^{2}v-2m(x)\Omega \dot{w}+m(x)\ddot{v}\\ -J_{\zeta \zeta }(x){\ddot{v}}^{\prime \prime }-J_{\zeta \zeta }{}^{\prime }(x){\ddot{v}}^{\prime }+J_{\eta \eta }(x)\Omega^{2}v^{\prime \prime }+J_{\eta \eta }{}^{\prime }(x)\Omega^{2}v^{\prime }=0 \end{array}$$
where the boundary condition at *x* = 0 are:(14)$$v(0)=0,\;w(0)=0,\;v^{\prime }(0)=0,\;w^{\prime }(0)=0$$
and at *x* = 1 are:(15)$$\begin{array}{c} I_{\eta \eta }(x)w^{\prime \prime \prime }+I_{\eta \eta }{}^{\prime }(x)w^{\prime \prime }-J_{\eta \eta }(x){\ddot{w}}^{\prime }+J_{\zeta \zeta }(x)\Omega^{2}w^{\prime }+M_{r}\Omega^{2}w-2M_{r}\Omega \dot{v}-M_{r}\ddot{w}=-\frac{\alpha_{w}V_{w}{}^{2}}{{\left(1-w\right)}^{2}}\left(1+\beta_{F}\frac{1-w}{b_{M}}\right)\\ I_{\zeta \zeta }(x)v^{\prime \prime \prime }+I_{\zeta \zeta }{}^{\prime }(x)v^{\prime \prime }-J_{\zeta \zeta }(x){\ddot{v}}^{\prime }+J_{\eta \eta }(x)\Omega^{2}v^{\prime }+M_{r}\Omega^{2}v+2M_{r}\Omega \dot{w}-M_{r}\ddot{v}=-\frac{\alpha_{v}V_{v}{}^{2}}{{\left(1-v\right)}^{2}}\left(1+\beta_{F}\frac{1-v}{h_{M}}\right)\\ v^{\prime \prime }(1)=0,w^{\prime \prime }(1)=0 \end{array}$$

The detailed expressions of the coefficients in Equations (13)–(15) can be found in Appendix A. The displacements *w*(*x*, *t*) and *v*(*x*, *t*) in Equations (13)–(15) can be expressed as:(16)$$w(x,t)=w_{s}(x)+w_{d}(x,t),\;v(x,t)=v_{s}(x)+v_{d}(x,t)$$

Substituting Equation (16) into Equations (13)–(15), the electrostatic force term is retained to order 3 by Taylor expansion. The expanded electrostatic force expressions can be found in Appendix B.

The main parameters of the micro-gyroscope were selected as follows [44]:(17)$$\begin{array}{c} L=4\times 10^{-4}\;m;d_{v}=d_{w}=2\times 10^{-6}\;m;E=1.6\times 10^{11}\;{N/m}^{2};h_{M}=12\times 10^{-6}\;m;\\ b_{M}=25\times 10^{-6}\;m;\beta_{F}=0.65;\rho =2300\;{\mathrm{kg}/m}^{3};\varepsilon_{0}=8.854\times 10^{-12}\;F/m;\\ M=7.2128\times 10^{-12}\;\mathrm{kg};b_{0}=a_{0}=2.8\times 10^{-6}\;m;A_{v}=A_{w}=3.92\times 10^{-12}\;m^{2}. \end{array}$$

## 3. Static Deflection and Pull-In Instability Analysis

If the bias DC voltage applied in the micro-gyroscope system reaches the critical value, pull-in instability will occur in the system [47]; therefore, it is necessary to carry out the static analysis of the micro-gyroscope system.

Substituting Equation (16) into Equations (13)–(15) and removing the time-dependent term, the static equations for the drive and sense directions are obtained, respectively. As the two equations in the drive and sense directions are symmetrical to each other, the following static analysis is carried out in the sense direction as an example:(18)$$I_{\zeta \zeta }(x)v_{s}^{\left(4\right)}+2I_{\zeta \zeta }{}^{\prime }(x)v_{s}^{\prime \prime \prime }+I_{\zeta \zeta }{}^{\prime \prime }(x)v_{s}^{\prime \prime }-m(x)\Omega^{2}v_{s}+J_{\eta \eta }(x)\Omega^{2}v_{s}^{\prime \prime }+J_{\eta \eta }{}^{\prime }(x)\Omega^{2}v_{s}^{\prime }=0$$
where the boundary condition at *x* = 0 are:(19)$$v_{s}(0)=0,v_{s}^{\prime }(0)=0$$
and at *x* = 1 are:(20)$$\begin{array}{c} I_{\zeta \zeta }(x)v_{s}^{\prime \prime \prime }+I_{\zeta \zeta }{}^{\prime }(x)v_{s}^{\prime \prime }+J_{\eta \eta }(x)\Omega^{2}v_{s}^{\prime }+M_{r}\Omega^{2}v_{s}=-\frac{\alpha_{v}V_{v}{}^{2}}{{\left(1-v_{s}\right)}^{2}}\left(1+\beta_{F}\frac{1-v_{s}}{h_{M}}\right)\\ v_{s}^{\prime \prime }(1)=0 \end{array}$$

In this work, DQM [48] is used to solve Equation (18). The one-dimensional function *v_s_* (*x*) is continuously derivable in the interval [0, 1] and *n* different nodes are taken in the interval, as shown in Figure 3. The *r*-order derivative of *v_s_* (*x*) at the discrete point $x_{i}(i=1,2,\ldots ,n)$ in the *x* direction of the grid is given by:
(21)$${v_{s}{}^{\left(r\right)}(x)|}_{x_{i}}={\frac{d^{r}v_{s}(x)}{dx^{r}}|}_{x_{i}}=\sum_{j=1}^{n}A_{i,j}^{\left(r\right)}v_{s}(x_{j}),\;i,j=1,2,\ldots ,n$$
where $A_{i,j}^{\left(r\right)}$ represents the coefficient of the *r*-order derivative. The grid point *x_j_* can be expressed as:(22)$$x_{j}=\frac{1}{2}\left[1-\cos\left(\frac{j-1}{n-1}\pi \right)\right]$$

The coefficients are given by the following recursive equations:(23)$$\begin{array}{ll} A_{i,j}^{\left(1\right)}=\frac{\prod_{v=1}^{n}(x_{i}-x_{v})}{\left(x_{i}-x_{j})\prod_{v=1}^{n}(x_{j}-x_{v}\right)}, & i,j=1,2,\cdots n,i\neq v,j\neq v\\ A_{i,j}^{\left(r\right)}=r\left(A_{i,i}{}^{\left(r-1\right)}A_{i,j}{}^{\left(1\right)}-\frac{A_{i,j}^{\left(r-1\right)}}{\left(x_{i}-x_{j}\right)}\right),\; & i,j=1,2,\cdots n\\ A_{i,i}^{\left(1\right)}=-\sum_{j=1,j\neq i}^{n}A_{i,j}^{\left(r\right)}, & i,j=1,2,\cdots n \end{array}$$

DQM is applied to Equation (18) and the boundary conditions of the micro-gyroscope. Here, we use the following form of the residual equation:(24)$$\begin{array}{c} r_{i}=I_{\zeta \zeta }(x_{i})\sum_{j=1}^{n}A_{i,j}^{\left(4\right)}v_{s,j}+2I_{\zeta \zeta }{}^{\prime }(x_{i})\sum_{j=1}^{n}A_{i,j}^{\left(3\right)}v_{s,j}+I_{\zeta \zeta }{}^{\prime \prime }(x_{i})\sum_{j=1}^{n}A_{i,j}^{\left(2\right)}v_{s,j}-m(x_{i})\Omega^{2}\sum_{j=1}^{n}A_{i,j}^{\left(0\right)}v_{s,j}\\ +J_{\eta \eta }(x_{i})\Omega^{2}\sum_{j=1}^{n}A_{i,j}^{\left(2\right)}v_{s,j}+J_{\eta \eta }{}^{\prime }(x_{i})\Omega^{2}\sum_{j=1}^{n}A_{i,j}^{\left(1\right)}v_{s,j}=0\;i=3,\ldots ,n-2 \end{array}$$
where the boundary conditions at *x* = 0 are:(25)$$r_{1}=\sum_{j=1}^{n}A_{1,j}^{\left(0\right)}v_{s,j}=0,\;r_{2}=\sum_{j=1}^{n}A_{1,j}^{\left(1\right)}v_{s,j}=0,\;i=1,j=1,\ldots ,n$$
and at *x* = 1 are:(26)$$\begin{array}{c} r_{n-1}=\sum_{j=1}^{n}A_{n,j}^{\left(2\right)}v_{j}=0,\;i=n,j=1,\ldots ,n\\ r_{n}=I_{\zeta \zeta }(x_{n})\sum_{j=1}^{n}A_{n,j}^{\left(3\right)}v_{s,j}+I_{\zeta \zeta }{}^{\prime }(x_{n})\sum_{j=1}^{n}A_{n,j}^{\left(2\right)}v_{s,j}+J_{\eta \eta }(x_{n})\Omega^{2}\sum_{j=1}^{n}A_{n,j}^{\left(1\right)}v_{s,j}\;+M_{r}\Omega^{2}\sum_{j=1}^{n}A_{n,j}^{\left(0\right)}v_{s,j}\\ =-\frac{\alpha_{v}V_{v}{}^{2}}{{\left(1-v_{s,n}\right)}^{2}}\left(1+\beta_{F}\frac{1-v_{s,n}}{h_{M}}\right)\sum_{j=1}^{n}A_{n,j}^{\left(0\right)} \end{array}$$
where $A^{\left(0\right)}$ represents the unit matrix of n×n, and $A_{i,j}^{\left(0\right)}$ represents the elements of the *i*-th row and *j*-th column of the unit matrix. Define the residual vector as $\vec{R}={\left[\begin{array}{ccccc} r_{1} & r_{2} & \cdots & r_{n-1} & r_{n} \end{array}\right]}^{T}$, where $r_{i}(i=1,\cdots ,n)$ is in the form shown in Equations (24)–(26).

Taking the derivative of the above residual equation for *v_s_*, Equations (24)–(26) can be written in the form of matrices as:(27)$$\left[\begin{array}{ccccc} A_{1,1}^{\left(0\right)} & A_{1,2}^{\left(0\right)} & \cdots & A_{1,n-1}^{\left(0\right)} & A_{1,n}^{\left(0\right)}\\ A_{1,1}^{\left(1\right)} & A_{1,2}^{\left(1\right)} & \cdots & A_{1,n-1}^{\left(1\right)} & A_{1,n}^{\left(1\right)}\\ A_{3,1} & A_{3,2} & \cdots & A_{3,n-1} & A_{4,n}\\ \vdots & \vdots & \vdots & \vdots & \vdots \\ A_{n-2,1} & A_{n-2,2} & \cdots & A_{n-2,n-1} & A_{n-2,n}\\ A_{n,1}^{\left(2\right)} & A_{n,2}^{\left(2\right)} & \cdots & A_{n,n-1}^{\left(2\right)} & A_{n,n}^{\left(2\right)}\\ A_{n,1} & A_{n,2} & \cdots & A_{n,n-1} & A_{n,n} \end{array}\right]$$
where the 1st and 2nd rows of the matrix (27) correspond to the boundary conditions (25), the 3rd to (*n* − 2)-th rows correspond to the Equation (24), and the (*n* − 1)-th and *n*-th rows correspond to the boundary conditions (26).

The expression of the element $A_{i,j}$ in the 3rd to (*n* − 2)-th rows of the matrix (27) is given by:(28)$$\begin{array}{c} A_{i,j}=I_{\zeta \zeta }(x_{i})A_{i,j}^{\left(4\right)}+2I_{\zeta \zeta }{}^{\prime }(x_{i})A_{i,j}^{\left(3\right)}+I_{\zeta \zeta }{}^{\prime \prime }(x_{i})A_{i,j}^{\left(2\right)}-m(x_{i})\Omega^{2}A_{i,j}^{\left(0\right)}+J_{\eta \eta }(x_{i})\Omega^{2}A_{i,j}^{\left(2\right)}+J_{\eta \eta }{}^{\prime }(x_{i})\Omega^{2}A_{i,j}^{\left(1\right)}\\ i=3,\ldots ,n-2 \end{array}$$
and the element $A_{n,j}$ of the *n*-th row is represented as follows:(29)$$\begin{array}{c} A_{n,j}=I_{\zeta \zeta }(x_{n})A_{n,j}^{\left(3\right)}+I_{\zeta \zeta }{}^{\prime }(x_{n})A_{n,j}^{\left(2\right)}+J_{\eta \eta }(x_{n})\Omega^{2}A_{n,j}^{\left(1\right)}+M_{r}\Omega^{2}A_{n,j}^{\left(0\right)}+\frac{\alpha_{v}V_{v}{}^{2}}{{\left(1-v_{s,n}\right)}^{2}}\left(1+\beta_{F}\frac{1-v_{s,n}}{h_{M}}\right)A_{n,j}^{\left(0\right)}\\ i=n,j=1,\ldots ,n \end{array}$$

The matrix (27) and the residual vector $\vec{R}$ are constructed into a set of equations, which are solved by Newton iterative method. The static deflection and pull-in voltage of the micro-gyroscope can be obtained.

The static deflection curves along the drive and sense directions are shown in Figure 4. The DC voltages *V_DC_*_1_ and *V_DC_*_2_ are selected as 3 V, the shape factor of width *α*_5_ = 0, and the shape factor of thickness *α*_4_ = −0.2, −0.1, 0, 0.1, …, 0.5. It can be found in Figure 4a that, as the shape factor *α*_4_ of the thickness increases, i.e., thickness *a*(*x*) decreases gradually along the beam length, the shape factor *α*_5_ of the width remains constant and, thus, the static deflection in the drive direction changes less. This is because the thickness variation primarily affects the stiffness of the tapered beam in the sense direction. Figure 4b shows the static deflection of the micro-gyroscope in the sense direction. It can be found that the static deflection in the sense direction increases significantly. The larger the shape factor, the smaller the bending stiffness of the beam, and the larger the static deflection of the tapered beam at the same voltage.

When the DC voltage exceeds the critical value (pull-in voltage), the stable equilibrium position of the microbeam no longer exists and the pull-in is triggered. To ensure the safe operating range of the micro-gyroscope, the pull-in voltage is critical to the stability of the system. The shape factors of the width and thickness are not equal, which results in different pull-in voltages in the drive and sense directions. As shown in Figure 5, the shape factors are selected as *α*_4_ = −0.2, −0.1, 0, 0.1, …, 0.5, and *α*_5_ = 0. The micro-gyroscope undergoes pull-in instability at 33% of the maximum displacement. It can be found in Figure 5a that when the shape factor *α*_4_ of the thickness gradually increases, the variation of the pull-in voltage in the drive direction is small. The pull-in voltages in the drive direction are 4.59 V, 4.74 V, 4.95 V, and 5.06 V, from left to right. However, as the shape factor *α*_4_ of the thickness increases, the pull-in voltage in the sense direction decreases significantly, as shown in Figure 5b. The pull-in voltages in the sense direction are 3.86 V, 4.11 V, 4.34 V, 4.54 V, 4.75 V, 4.95 V, 5.12 V, and 5.3 V, from left to right. This is mainly due to the fact that *α*_4_ affects the stiffness of the beam in the sense direction. Therefore, changing the shape factor of the width (thickness) mainly affects the pull-in voltage and static deflection in the drive (sense) direction. When *α*_4_ = *α*_5_ = 0, the static pull-in voltages in the drive and sense directions are equal, both being 4.95 V, which is consistent with the results in the literature [31]. The accuracy of the DQM and the established motion governing equations in this paper is further verified.

Therefore, in the subsequent studies (linear system analysis and effects of electrostatic force nonlinearity), none of the DC voltages we applied exceeded the pull-in voltage.

## 4. Study of Natural Frequency and Mode Shape of Micro-Gyroscope

As a result of the FM micro-gyroscope detecting the angular velocity mainly by the difference of the natural frequency, the natural frequency analysis is required. Expressing the dynamic component of the micro-gyroscope as a product of a spatial function and a temporal function, the displacements *v* and *w* can be expressed as follows:(30)$$v=v_{s}+(\varphi_{v}(x)e^{i\omega t}+cc),w=w_{s}+(\varphi_{w}(x)e^{i\omega t}+cc)$$
where $\varphi_{v}(x)$ and $\varphi_{w}(x)$ are the mode shape in the sense and drive directions, respectively. *cc* denotes the complex conjugate.

Substituting Equation (30) into Equations (13)–(15), the first term of the electrostatic force in equation (15) is retained. The static term in the obtained equation is omitted. Using DQM to analyze the natural frequency of the system, the governing equations of the micro-gyroscope can be expressed as:(31)$$\begin{array}{c} I_{\eta \eta }(x_{i})\sum_{j=1}^{n}A_{i,j}^{\left(4\right)}\varphi_{w}{}_{j}+2I_{\eta \eta }{}^{\prime }(x_{i})\sum_{j=1}^{n}A_{i,j}^{\left(3\right)}\varphi_{w}{}_{j}+I_{\eta \eta }{}^{\prime \prime }(x_{i})\sum_{j=1}^{n}A_{i,j}^{\left(2\right)}\varphi_{w}{}_{j}-m(x_{i})\Omega^{2}\sum_{j=1}^{n}A_{i,j}^{\left(0\right)}\varphi_{w}{}_{j}\\ +2i\omega m(x_{i})\Omega \sum_{j=1}^{n}A_{i,j}^{\left(0\right)}\varphi_{v}{}_{j}-\omega^{2}m(x_{i})\sum_{j=1}^{n}A_{i,j}^{\left(0\right)}\varphi_{w}{}_{j}+\omega^{2}J_{\eta \eta }(x_{i})\sum_{j=1}^{n}A_{i,j}^{\left(2\right)}\varphi_{w}{}_{j}+\omega^{2}J_{\eta \eta }{}^{\prime }(x_{i})\sum_{j=1}^{n}A_{i,j}^{\left(1\right)}\varphi_{w}{}_{j}\\ +J_{\zeta \zeta }(x_{i})\Omega^{2}\sum_{j=1}^{n}A_{i,j}^{\left(2\right)}\varphi_{w}{}_{j}+J_{\zeta \zeta }{}^{\prime }(x_{i})\Omega^{2}\sum_{j=1}^{n}A_{i,j}^{\left(1\right)}\varphi_{w}{}_{j}=0\\ i=3,4,\ldots ,n-2 \end{array}$$
(32)$$\begin{array}{c} I_{\zeta \zeta }(x_{i})\sum_{j=1}^{n}A_{i,j}^{\left(4\right)}\varphi_{v}{}_{j}+2I_{\zeta \zeta }{}^{\prime }(x_{i})\sum_{j=1}^{n}A_{i,j}^{\left(3\right)}\varphi_{v}{}_{j}+I_{\zeta \zeta }{}^{\prime \prime }(x_{i})\sum_{j=1}^{n}A_{i,j}^{\left(2\right)}\varphi_{v}{}_{j}-m(x_{i})\Omega^{2}\sum_{j=1}^{n}A_{i,j}^{\left(0\right)}\varphi_{v}{}_{j}\\ -2i\omega m(x_{i})\Omega \sum_{j=1}^{n}A_{i,j}^{\left(0\right)}\varphi_{w}{}_{j}-\omega^{2}m(x_{i})\sum_{j=1}^{n}A_{i,j}^{\left(0\right)}\varphi_{v}{}_{j}+\omega^{2}J_{\zeta \zeta }(x_{i})\sum_{j=1}^{n}A_{i,j}^{\left(2\right)}\varphi_{v}{}_{j}+\omega^{2}J_{\zeta \zeta }{}^{\prime }(x_{i})\sum_{j=1}^{n}A_{i,j}^{\left(1\right)}\varphi_{v}{}_{j}\\ +J_{\eta \eta }(x_{i})\Omega^{2}\sum_{j=1}^{n}A_{i,j}^{\left(2\right)}\varphi_{v}{}_{j}+J_{\eta \eta }{}^{\prime }(x_{i})\Omega^{2}\sum_{j=1}^{n}A_{i,j}^{\left(1\right)}\varphi_{v}{}_{j}=0\\ i=3,4,\ldots ,n-2 \end{array}$$
where the boundary condition at *x* = 0 are:(33)$$\sum_{j=1}^{n}A_{1,j}^{\left(0\right)}\varphi_{v}{}_{j}=0,\sum_{j=1}^{n}A_{1,j}^{\left(0\right)}\varphi_{w}{}_{j}=0,\sum_{j=1}^{n}A_{1,j}^{\left(1\right)}\varphi_{v}{}_{j}=0,\sum_{j=1}^{n}A_{1,j}^{\left(1\right)}\varphi_{w}{}_{j}=0,i=1,j=1,\ldots ,n$$
and at *x* = 1 are:(34)$$\begin{array}{c} I_{\eta \eta }(x_{n})\sum_{j=1}^{n}A_{n,j}^{\left(3\right)}\varphi_{w}{}_{j}+I_{\eta \eta }{}^{\prime }(x_{n})\sum_{j=1}^{n}A_{n,j}^{\left(2\right)}\varphi_{w}{}_{j}+\omega^{2}J_{\eta \eta }(x_{n})\sum_{j=1}^{n}A_{n,j}^{\left(1\right)}\varphi_{w}{}_{j}+J_{\zeta \zeta }(x_{n})\Omega^{2}\sum_{j=1}^{n}A_{n,j}^{\left(1\right)}\varphi_{w}{}_{j}\\ +M_{r}\Omega^{2}\sum_{j=1}^{n}A_{n,j}^{\left(0\right)}\varphi_{w}{}_{j}-2i\omega M_{r}\Omega \sum_{j=1}^{n}A_{n,j}^{\left(0\right)}\varphi_{v}{}_{j}+\omega^{2}M_{r}\sum_{j=1}^{n}A_{n,j}^{\left(0\right)}\varphi_{w}{}_{j}\\ =-\frac{\alpha_{w}V_{w}{}^{2}}{{\left(1-w_{s}\right)}^{3}}\left(2+\beta_{F}\frac{1-w_{s}}{b_{M}}\right)\sum_{j=1}^{n}A_{n,j}^{\left(0\right)}\varphi_{w}{}_{j}\\ I_{\zeta \zeta }(x_{n})\sum_{j=1}^{n}A_{n,j}^{\left(3\right)}\varphi_{v}{}_{j}+I_{\zeta \zeta }{}^{\prime }(x_{n})\sum_{j=1}^{n}A_{n,j}^{\left(2\right)}\varphi_{v}{}_{j}+\omega^{2}J_{\zeta \zeta }(x_{n})\sum_{j=1}^{n}A_{n,j}^{\left(1\right)}\varphi_{v}{}_{j}+J_{\eta \eta }(x_{n})\Omega^{2}\sum_{j=1}^{n}A_{n,j}^{\left(1\right)}\varphi_{v}{}_{j}\\ +M_{r}\Omega^{2}\sum_{j=1}^{n}A_{n,j}^{\left(0\right)}\varphi_{v}{}_{j}+2i\omega M_{r}\Omega \sum_{j=1}^{n}A_{n,j}^{\left(0\right)}\varphi_{w}{}_{j}+\omega^{2}M_{r}\sum_{j=1}^{n}A_{n,j}^{\left(0\right)}\varphi_{v}{}_{j}\\ =-\frac{\alpha_{v}V_{v}{}^{2}}{{\left(1-v_{s}\right)}^{3}}\left(2+\beta_{F}\frac{1-v_{s}}{h_{M}}\right)\sum_{j=1}^{n}A_{n,j}^{\left(0\right)}\varphi_{v}{}_{j}\\ \sum_{j=1}^{n}A_{n,j}^{\left(2\right)}\varphi_{v}{}_{j}=0,\sum_{j=1}^{n}A_{n,j}^{\left(2\right)}\varphi_{w}{}_{j}=0i=n,j=1,\ldots ,n \end{array}$$

Rewrite Equations (31)–(34) in the form of matrix as:(35)$$\left(\left[M]+\omega^{2}[K\right]\right)C=0$$
where *M* and *K* are 2n×2n matrices, $C={\left[\begin{array}{cccccccc} \varphi_{w}{}_{1} & \varphi_{w}{}_{2} & \cdots & \varphi_{w}{}_{n} & \varphi_{v}{}_{1} & \varphi_{v}{}_{2} & \cdots & \varphi_{v}{}_{n} \end{array}\right]}^{T}$. By making the determinant $\left|\left[M]+\omega^{2}[K\right]\right|$ equal to zero, the natural frequency of the system and the corresponding mode shape can be obtained.

To verify the accuracy of the natural frequency using DQM, the convergence analysis of the natural frequency is first performed. The parameters shown in Equation (14) are selected and the DC voltage in the drive and sense directions is 0 V. Figure 6 gives the natural frequency and the first three orders mode shapes of the system for the different grid points. It can be found that the first-order natural frequency of the tapered beam gradually converges and stabilizes as the number of grid points increases to 12. Therefore, the number of grid points is selected to be 12 for the following natural frequency analysis.

Figure 7 shows the relationship between the natural frequency and the shape factors of the tapered beam. The shape factor of the width is equal to that of the thickness; that is, *α*_4_ = *α*_5_. *V_DC_*_1_ = *V_DC_*_2_= 0 V. As the shape factor of the tapered beam increases gradually, the first-order natural frequency of the system decreases. Therefore, the natural frequency of the micro-gyroscope can be adjusted by the shape factors to meet the design requirements. The theoretical results are in good agreement with the finite element simulation (FES), which further verifies the accuracy of the governing equations.

The relationship between the first-order natural frequency of the micro-gyroscope and the angular velocity is shown in Figure 8. Due to the presence of gyroscopic effects in the rotating structure, the natural frequencies of the drive and sense modes are caused to vary with the angular velocity. As can be seen from Figure 8, in a wide range of angular velocity, the natural frequency of the micro-gyroscope system with shape factors of 0 and 0.5 changes linearly with the increase in the angular velocity. One of the natural frequencies increases with the increase in the angular velocity, while the other decreases. The FM method uses the principle of frequency separation due to the gyroscopic effect, and the angular velocity can be obtained by measuring the difference between the natural frequencies of the system.

## 5. Linear Analysis of the Micro-Gyroscope System

In the previous section, DQM was used to calculate the natural frequencies of the micro-gyroscope system. It can be found that, starting from the base rotation, the natural frequencies gradually separate into a pair of spaced natural frequencies. To investigate the influences of the system parameters on the performance of the FM micro-gyroscope, the linear analysis in this section is carried out first.

As the electrostatic force acts on the tip mass, the work *W_w_* and *V_v_* conducted by the electrostatic field in the drive and sense directions can be expressed as a Dirac delta function in the form of:(36)$$\begin{array}{l} \delta W_{v}=\int_{0}^{L}\delta (x-L)\frac{\varepsilon_{0}A_{v}V_{v}{}^{2}}{2{\left(d_{v}-v\right)}^{2}}\left(1+\beta_{F}\frac{d_{v}-v}{h_{M}}\right)\delta vdx,\\ \delta W_{w}=\int_{0}^{L}\delta (x-L)\frac{\varepsilon_{0}A_{w}V_{w}{}^{2}}{2{\left(d_{w}-w\right)}^{2}}\left(1+\beta_{F}\frac{d_{w}-w}{b_{M}}\right)\delta wdx \end{array}$$

Therefore, the electrostatic force term can be converted from the boundary conditions into the governing equations. Galerkin discretization is performed on Equation (13) and the vibration displacements of the micro-gyroscope in the drive and sense directions can be expressed as:(37)$$w(x,t)=\varphi_{wr}(x)p_{r}(t),\;v(x,t)=\varphi_{vr}(x)q_{r}(t)$$
where $\varphi_{wr}(x)$ and $\varphi_{vr}(x)$ are the *r*-th order mode shape in the drive and sense directions, respectively. *q_r_*(*t*) and *p_r_*(*t*) are the *r*-th order generalized coordinates. The first-order mode shape of the designed micro-gyroscope is the operating mode, a single-mode approximation is adopted [49], that is, *r* is set to 1.

The first equation of Equation (13) is multiplied by $\varphi_{w1}(x)$ and the second equation is multiplied by $\varphi_{v1}(x)$, then the two equations are integrated from 0 to 1. The discretized reduced-order model (ROM) in the drive and sense directions can be obtained as:(38)$$\begin{array}{c} \left(\int_{0}^{1}m(x)\varphi_{w1}(x)\varphi_{w1}(x)dx-\int_{0}^{1}J_{\eta \eta }(x)\varphi_{w1}{}^{\prime \prime }(x)\varphi_{w1}(x)dx-\int_{0}^{1}J_{\eta \eta }{}^{\prime }(x)\varphi_{w1}{}^{\prime }(x)\varphi_{w1}(x)dx\right)\ddot{p}(t)\\ +\left(\begin{array}{c} \int_{0}^{1}I_{\eta \eta }(x)\varphi_{w1}{}^{\left(4\right)}(x)\varphi_{w1}(x)dx+\int_{0}^{1}2I_{\eta \eta }{}^{\prime }(x)\varphi_{w1}{}^{\prime \prime \prime }(x)\varphi_{w1}(x)dx\\ +\int_{0}^{1}I_{\eta \eta }{}^{\prime \prime }(x)\varphi_{w1}{}^{\prime \prime }(x)\varphi_{w1}(x)dx-\Omega^{2}\int_{0}^{1}m(x)\varphi_{w1}(x)\varphi_{w1}(x)dx\\ +\Omega^{2}\int_{0}^{1}J_{\zeta \zeta }(x)\varphi_{w1}{}^{\prime \prime }(x)\varphi_{w1}(x)dx+\Omega^{2}\int_{0}^{1}J_{\zeta \zeta }{}^{\prime }(x)\varphi_{w1}{}^{\prime }(x)\varphi_{w1}(x)dx \end{array}\right)p(t)\\ +c\int_{0}^{1}\varphi_{w1}(x)\varphi_{w1}(x)dx\dot{p}(t)+2\Omega \int_{0}^{1}m(x)\varphi_{v1}(x)\varphi_{w1}(x)dx\dot{q}(t)=a_{w}{\left(V_{DC1}+V_{AC}\cos\left(\omega_{0}t\right)\right)}^{2}\int_{0}^{1}F(x)\varphi_{w1}(x)dx \end{array}$$
(39)$$\begin{array}{c} \left(\int_{0}^{1}m(x)\varphi_{v1}(x)\varphi_{v1}(x)dx-\int_{0}^{1}J_{\zeta \zeta }(x)\varphi_{v1}{}^{\prime \prime }(x)\varphi_{v1}(x)dx-\int_{0}^{1}J_{\zeta \zeta }{}^{\prime }(x)\varphi_{v1}{}^{\prime }(x)\varphi_{v1}(x)dx\right)\ddot{q}(t)\\ +\left(\begin{array}{l} \int_{0}^{1}I_{\zeta \zeta }(x)\varphi_{v1}{}^{\left(4\right)}(x)\varphi_{v1}(x)dx+\int_{0}^{1}2I_{\zeta \zeta }{}^{\prime }(x)\varphi_{v1}{}^{\prime \prime \prime }(x)\varphi_{v1}(x)dx\\ +\int_{0}^{1}I_{\zeta \zeta }{}^{\prime \prime }(x)\varphi_{v1}{}^{\prime \prime }(x)\varphi_{v1}(x)dx-\Omega^{2}\int_{0}^{1}m(x)\varphi_{v1}(x)\varphi_{v1}(x)dx\\ +\Omega^{2}\int_{0}^{1}J_{\eta \eta }(x)\varphi_{v1}{}^{\prime \prime }(x)\varphi_{v1}(x)dx+\Omega^{2}\int_{0}^{1}J_{\eta \eta }{}^{\prime }(x)\varphi_{v1}{}^{\prime }(x)\varphi_{v1}(x)dx \end{array}\right)q(t)\\ +c\int_{0}^{1}\varphi_{v1}(x)\varphi_{v1}(x)dx\dot{q}(t)-2\Omega \int_{0}^{1}m(x)\varphi_{w1}(x)\varphi_{v1}(x)dx\dot{p}(t)=a_{v}V_{DC2}{}^{2}\int_{0}^{1}G(x)\varphi_{v1}(x)dx \end{array}$$
where *G*(*x*) and *F*(*x*) are the electrostatic force term.

Due to the mode shapes being obtained by numerical methods, they are represented here by the fitting functions. To ensure that the fitting function has a high enough accuracy, it can be substituted into the ROM. By comparing the natural frequencies of the ROM system with those of the original system, it is necessary to verify whether the fitting function meets the accuracy requirement.

The mode shape functions of the drive and sense modes are as follows:(40)$$\begin{array}{l} \varphi_{w}(x)=a_{0}+a_{1}x+a_{2}x^{2}+\cdots \cdots +a_{n-1}x^{n_{2}-1}+a_{n}x^{n_{2}}\\ \varphi_{v}(x)=b_{0}+b_{1}x+b_{2}x^{2}+\cdots \cdots +b_{n-1}x^{n_{2}-1}+b_{n}x^{n_{2}} \end{array}$$
where *n*_2_ is the highest power of the fitting function.

Table 1 gives the natural frequencies of the ROM system at different highest powers. Comparing the natural frequency in the ROM system with the natural frequency of the original system, it can be found that, with the increase in the highest power of the fitting function, the errors between them decrease. As shown in Table 1, for *α*_4_ = *α*_5_ = 0, *n*_2_ = 8; for *α*_4_ = 0 and *α*_5_ = 0.5, *n*_2_ = 16 in the drive direction and *n*_2_ = 14 in the sense direction; for *α*_4_ = 0.5 and *α*_5_ = 0.5, *n*_2_ = 16, the mode shape function has high precision.

Next, the linear analysis of the FM micro-gyroscope is performed. Neglecting the damping, external excitation, and electrostatic force nonlinearity in Equations (38) and (39), the second-order ordinary differential equation can be obtained as:(41)$$M\ddot{Q}+G\dot{Q}+KQ=0$$
where
(42)$$Q=\left[\begin{array}{c} p\\ q \end{array}\right],M=\left[\begin{array}{cc} 1 & 0\\ 0 & 1 \end{array}\right],G=\left[\begin{array}{cc} 0 & c_{1}\Omega \\ -c_{3}\Omega & 0 \end{array}\right],K=\left[\begin{array}{cc} \omega_{1}{}^{2}+c_{2}\Omega^{2} & 0\\ 0 & \omega_{2}{}^{2}+c_{4}\Omega^{2} \end{array}\right]$$
and the coefficients of Equation (41) can be found in Appendix C.

Substituting $Q=\mu e^{i\omega t}$ into Equation (41), the natural frequencies of the micro-gyroscope system can be obtained as:(43)$$\begin{array}{l} \omega_{n1}=\frac{\sqrt{\omega_{1}{}^{2}+\omega_{2}{}^{2}+c_{1}c_{3}\Omega^{2}+c_{2}\Omega^{2}+c_{4}\Omega^{2}-\sqrt{-4\left(\omega_{1}{}^{2}+c_{2}\Omega^{2}\right)\left(\omega_{2}{}^{2}+c_{4}\Omega^{2}\right)+{\left(\omega_{1}{}^{2}+\omega_{2}{}^{2}+c_{1}c_{3}\Omega^{2}+c_{2}\Omega^{2}+c_{4}\Omega^{2}\right)}^{2}}}}{\sqrt{2}}\\ \omega_{n2}=\frac{\sqrt{\omega_{1}{}^{2}+\omega_{2}{}^{2}+c_{1}c_{3}\Omega^{2}+c_{2}\Omega^{2}+c_{4}\Omega^{2}+\sqrt{-4\left(\omega_{1}{}^{2}+c_{2}\Omega^{2}\right)\left(\omega_{2}{}^{2}+c_{4}\Omega^{2}\right)+{\left(\omega_{1}{}^{2}+\omega_{2}{}^{2}+c_{1}c_{3}\Omega^{2}+c_{2}\Omega^{2}+c_{4}\Omega^{2}\right)}^{2}}}}{\sqrt{2}} \end{array}$$
where *ω_n_*_1_ and *ω_n_*_2_ are the frequencies that gradually decrease and increase with increasing angle velocity, respectively.

When the shape factors of the width and thickness are equal, the cross-section of the tapered beam is always square and we have *ω*_1_ = *ω*_2_, *c*_1_ = *c*_3_, *c*_2_ = *c*_4_. The relationship between the frequency difference of the system and the angular velocity can be obtained as:(44)$$\omega_{n2}-\omega_{n1}=\Delta \omega =c_{1}\Omega =2\kappa \Omega$$
(45)$$\kappa =\frac{\int_{0}^{1}m(x)\varphi_{v1}(x)\varphi_{w1}(x)dx}{\int_{0}^{1}m(x)\varphi_{w1}(x)\varphi_{w1}(x)dx-\int_{0}^{1}J_{\eta \eta }(x)\varphi_{w1}{}^{\prime \prime }(x)\varphi_{w1}(x)dx-\int_{0}^{1}J_{\eta \eta }{}^{\prime }(x)\varphi_{w1}{}^{\prime }(x)\varphi_{w1}(x)dx}$$

Assuming *α*_4_ = *α*_5_, the relationship between the first-order frequency and the angular velocity for different shape factors is shown in Figure 9. The shape factors are −0.3, 0, 0.3, and 0.5, respectively. As shown in Figure 9a, for different shape factors, the trend of the natural frequency with the angular velocity is the same. The intersection of *ω_n_*_1_ and the horizontal axis is defined as the critical point. With the angular velocity increasing gradually, the frequency *ω_n_*_2_ increases, the frequency *ω_n_*_1_ decreases first, and when the critical point is exceeded, *ω_n_*_1_ starts to increase. The smaller the shape factor, the greater the structural stiffness of the tapered beam and, thus, the higher the natural frequency and critical point. Figure 9b shows the variation of the natural frequency difference with the angular velocity and the slope represents the sensitivity of the FM micro-gyroscope. It can be found that the shape factor of the tapered beam has little effect on the sensitivity. When the critical point is exceeded, the natural frequency difference no longer changes with the angular velocity and it is not suitable to detect the angular velocity. Therefore, the higher the critical point value, the greater the dynamic range of the micro-gyroscope.

The detection performance of the FM micro-gyroscope with width and thickness asymmetry is given in Figure 10. As the difference between the width and thickness shape factors increases, the initial mismatch of the natural frequencies increases gradually. Figure 10a shows that when $\alpha_{4}\neq \alpha_{5}$, the angular velocity changes nonlinearly in a small range and then linearly. The larger the difference between the shape factors of the thickness and width, the smaller the sensitivity. At larger angular velocities, the difference in the natural frequencies of the two modes varies linearly with the angular velocity, but the sensitivity is always less than that of the same shape factors, as shown in Figure 10b and Table 2. In practice, due to the structural design or machining errors, a micro-gyroscope may have a different width and thickness of the tapered beam, resulting in a reduced FM sensitivity performance. Therefore, structural asymmetry should be avoided as much as possible.

Figure 11 shows the effects of the DC voltage in the drive and sense directions on the performance of the FM micro-gyroscope. The system parameters are set as: *α*_4_ = *α*_5_ = 0, black curve represents *V_DC_*_1_ = *V_DC_*_2_ = 2.5 V; red curve represents *V_DC_*_1_ = 3 V, *V_DC_*_2_ = 2 V; blue curve represents *V_DC_*_1_ = 3.5 V, *V_DC_*_2_ = 1.5 V. As shown in Figure 11a, it can be found that, with the increase in the DC voltage difference between the drive and sense directions, the mismatch between the natural frequencies of the two modes increases. When the measured angular velocity is less than 1000 rad/s, the natural frequency of the system changes slowly, as shown in Figure 11b. The original linear system is destroyed, and the measurement accuracy of the micro-gyroscope is reduced. It is unsuitable for measuring angular velocity at this time. As the angular velocity increases, the system sensitivity gradually returns to a linear variation. At this point, the system sensitivity is the same as when the voltage is equal in both directions. In order to ensure the sensitivity of the FM micro-gyroscope, the DC voltage applied in the drive and sense directions should be equal.

If the shape factors and DC voltages in the drive and sense directions are not equal, they will lead to unequal natural frequencies in the drive and sense directions, thus reducing the sensitivity of the FM micro-gyroscope. Therefore, if the beam micro-gyroscope has a rectangular cross-section, we can apply different DC voltages in the drive and sense directions to perform mode-matching again, which will also improve the FM performance of the micro-gyroscope.

## 6. Influence of Electrostatic Force Nonlinearity on the Sensitivity of FM Micro-Gyroscope

The electrostatic force nonlinearity is common in micro-gyroscopes, which affects the characteristics of the system, such as the natural frequency and dynamic response. It is important to investigate the influence of nonlinear factors on the FM micro-gyroscope. As it is difficult to continue the analysis using DQM after considering the electrostatic nonlinearity, in this section, the effects of the angular velocity and initial value of the micro-gyroscope system on the nonlinear natural frequency are investigated by using the IMM.

Removing the damping term and AC voltage term in Equations (38) and (39), we obtain the following equation:(46)$$\ddot{p}(t)+\omega_{1}{}^{2}p(t)+c_{11}\dot{q}(t)+c_{21}p{\left(t\right)}^{2}+c_{31}p{\left(t\right)}^{3}=0$$
(47)$$\ddot{q}(t)+\omega_{2}{}^{2}q(t)-c_{12}\dot{p}(t)+c_{22}q{\left(t\right)}^{2}+c_{32}q{\left(t\right)}^{3}=0$$
where the coefficients of Equations (46) and (47) can be found in Appendix D.

Converting the modal coordinates in Equations (46) and (47) into state vectors:(48)$$\left[\begin{array}{cccc} x_{1} & y_{1} & x_{2} & y_{2} \end{array}\right]=\left[\begin{array}{cccc} p & \dot{p} & q & \dot{q} \end{array}\right]$$

Then, the motion equations can be written as:(49)$$\begin{array}{l} {\dot{x}}_{1}=y_{1},\;{\dot{y}}_{1}=-\omega_{1}{}^{2}x_{1}-c_{11}{\dot{x}}_{2}-c_{21}x_{1}{}^{2}-c_{31}x_{1}{}^{3}\\ {\dot{x}}_{2}=y_{2},\;{\dot{y}}_{2}=-\omega_{2}{}^{2}x_{2}+c_{12}{\dot{x}}_{1}-c_{22}x_{2}{}^{2}-c_{32}x_{2}{}^{3} \end{array}$$

According to the IMM, the state vectors *x*_1_, *x*_2_, *y*_1_ and *y*_2_ related to nonlinearity can be expressed as:(50)$$\begin{array}{l} x_{1}=u,\;y_{1}=v\\ x_{2}=a_{1}u+a_{2}v+a_{3}u^{2}+a_{4}v^{2}+a_{5}u^{3}+a_{6}v^{3}\\ y_{2}=b_{1}u+b_{2}v+b_{3}u^{3}+b_{4}v^{4}+b_{5}u^{3}+b_{6}v^{3} \end{array}$$

The derivatives of *x*_2_ and *y*_2_ are:(51)$$\begin{array}{l} {\dot{x}}_{2}=a_{1}v+a_{2}\dot{v}+2a_{3}uv+2a_{4}v\dot{v}+3a_{5}u^{2}v+3a_{6}v^{2}\dot{v}\\ {\dot{y}}_{2}=b_{1}v+b_{2}\dot{v}+2b_{3}uv+2b_{4}v\dot{v}+3b_{5}u^{2}v+3b_{6}v^{2}\dot{v} \end{array}$$
(52)$$\begin{array}{l} {\dot{x}}_{2}=y_{2}=b_{1}u+b_{2}v+b_{3}u^{3}+b_{4}v^{4}+b_{5}u^{3}+b_{6}v^{3}\\ \dot{v}={\dot{y}}_{1}=-\omega_{1}{}^{2}x_{1}-c_{11}{\dot{x}}_{2}-c_{21}x_{1}{}^{2}-c_{31}x_{1}{}^{3}=-\omega_{1}{}^{2}u-c_{11}y_{2}-c_{21}u^{2}-c_{31}u^{3}\\ {\dot{y}}_{2}=-\omega_{2}{}^{2}x_{2}+c_{12}{\dot{x}}_{1}-c_{22}x_{2}{}^{2}-c_{32}x_{2}{}^{3}=-\omega_{2}{}^{2}x_{2}+c_{12}v-c_{22}x_{2}{}^{2}-c_{32}x_{2}{}^{3}\\ x_{2}=a_{1}u+a_{2}v+a_{3}u^{2}+a_{4}v^{2}+a_{5}u^{3}+a_{6}v^{3} \end{array}$$

Substituting Equations (49), (50), and (52) into (51), the final equations for *u* and *v* can be obtained as follows:(53)$$\begin{array}{l} b_{1}u+b_{2}v+b_{3}u^{2}+b_{4}v^{2}+b_{5}u^{3}+b_{6}v^{3}=a_{1}v+2a_{3}uv+3a_{5}u^{2}v+\\ a_{2}\left(-\omega_{1}{}^{2}u-\left(b_{1}u+b_{2}v+b_{3}u^{2}+b_{4}v^{2}+b_{5}u^{3}+b_{6}v^{3}\right)c_{11}-c_{21}u^{2}-c_{31}u^{3}\right)+\\ 2a_{4}v\left(-\omega_{1}{}^{2}u-\left(b_{1}u+b_{2}v+b_{3}u^{2}+b_{4}v^{2}+b_{5}u^{3}+b_{6}v^{3}\right)c_{11}-c_{21}u^{2}-c_{31}u^{3}\right)\\ +3a_{6}v^{2}\left(-\omega_{1}{}^{2}u-\left(b_{1}u+b_{2}v+b_{3}u^{2}+b_{4}v^{2}+b_{5}u^{3}+b_{6}v^{3}\right)c_{11}-c_{21}u^{2}-c_{31}u^{3}\right)=0 \end{array}$$
(54)$$\begin{array}{l} -c_{12}v+\omega_{2}{}^{2}\left(a_{1}u+a_{2}v+a_{3}u^{2}+a_{4}v^{2}+a_{5}u^{3}+a_{6}v^{3}\right)+b_{1}v+2b_{3}uv+3b_{5}u^{2}v+\\ {\left(a_{1}u+a_{2}v+a_{3}u^{2}+a_{4}v^{2}+a_{5}u^{3}+a_{6}v^{3}\right)}^{2}c_{22}+{\left(a_{1}u+a_{2}v+a_{3}u^{2}+a_{4}v^{2}+a_{5}u^{3}+a_{6}v^{3}\right)}^{3}c_{32}+\\ b_{2}\left(-\omega_{1}{}^{2}u-\left(b_{1}u+b_{2}v+b_{3}u^{2}+b_{4}v^{2}+b_{5}u^{3}+b_{6}v^{3}\right)c_{11}-c_{21}u^{2}-c_{31}u^{3}\right)\\ +2b_{4}v\left(-\omega_{1}{}^{2}u-\left(b_{1}u+b_{2}v+b_{3}u^{2}+b_{4}v^{2}+b_{5}u^{3}+b_{6}v^{3}\right)c_{11}-c_{21}u^{2}-c_{31}u^{3}\right)+\\ 3b_{6}v^{2}\left(-\omega_{1}{}^{2}u-\left(b_{1}u+b_{2}v+b_{3}u^{2}+b_{4}v^{2}+b_{5}u^{3}+b_{6}v^{3}\right)c_{11}-c_{21}u^{2}-c_{31}u^{3}\right)=0 \end{array}$$
where Equations (53) and (54) have all linear and nonlinear terms for the variables *u*, *v*, *u*^2^, *v*^2^, *u*^3^, and *v*^3^. The values of these unknowns (*a_k_* and *b_k_*, *k* = 1, …, 6) can be obtained by making the coefficient of similar terms equal to zero. A series of nonlinear algebraic equations with relevant variables as *u*, *v*, *u*^2^, *v*^2^, *u*^3^, and *v*^3^ are:(55)$$\begin{array}{ll} u\;term: & \omega_{1}{}^{2}a_{2}+b_{1}+a_{2}b_{1}c_{11}=0\\ & \omega_{2}{}^{2}a_{1}-\omega_{1}{}^{2}b_{2}-b_{1}b_{2}c_{11}=0\\ v\;term: & a_{1}-b_{2}-a_{2}b_{2}c_{11}=0\\ & \omega_{2}{}^{2}a_{2}-c12+b_{1}-b_{2}^{2}c_{11}=0\\ u^{2}\;term: & b_{3}+a_{2}b_{3}c_{11}+a_{2}c_{21}=0\\ & \omega_{2}{}^{2}a_{3}-b_{2}b_{3}c_{11}-b_{2}c_{21}+a_{1}^{2}c_{22}=0\\ v^{2}\;term: & b_{4}+2a_{4}b_{2}c_{11}+a_{2}b_{4}c_{11}=0\\ & \omega_{2}{}^{2}a_{4}-3b_{2}b_{4}c_{11}+a_{2}^{2}c_{22}=0\\ u^{3}\;term: & -b_{5}-a_{2}b_{5}c_{11}-a_{2}c_{31}=0\\ & \omega_{2}{}^{2}a_{5}-b_{2}b_{5}c_{11}+2a_{1}a_{3}c_{22}-b_{2}c_{31}+a_{1}^{3}c_{32}=0\\ v^{3}\;term: & -b_{6}-3a_{6}b_{2}c_{11}-2a_{4}b_{4}c_{11}-a_{2}b_{6}c_{11}=0\\ & \omega_{2}{}^{2}a_{6}-2b_{4}^{2}c_{11}-4b_{2}b_{6}c_{11}+2a_{2}a_{4}c_{22}+a_{2}^{3}c_{32}=0 \end{array}$$

The two sets of real roots of *a_k_* and *b_k_* can be solved. Mode 1:(56)$$\begin{array}{l} a_{1}=a_{3}=a_{5}=b_{2}=b_{4}=b_{6}=0,\\ a_{2}=\frac{\omega_{1}{}^{2}-\omega_{2}{}^{2}+c_{12}c_{11}-\sqrt{4\omega_{2}{}^{2}c_{11}c_{12}+{\left(\omega_{1}{}^{2}-\omega_{2}{}^{2}+c_{12}c_{11}\right)}^{2}}}{2\omega_{2}{}^{2}c_{11}},\\ a_{4}=-\frac{{\left(\omega_{1}{}^{2}-\omega_{2}{}^{2}+c_{12}c_{11}-\sqrt{4\omega_{2}{}^{2}c_{12}c_{11}+{\left(\omega_{1}{}^{2}-\omega_{2}{}^{2}+c_{12}c_{11}\right)}^{2}}\right)}^{2}c_{22}}{4\omega_{2}{}^{6}c_{11}^{2}},\\ a_{6}=\frac{{\left(-\omega_{1}{}^{2}+\omega_{2}{}^{2}-c_{12}c_{11}+\sqrt{4c_{12}\omega_{2}{}^{2}c_{11}+{\left(\omega_{1}{}^{2}-\omega_{2}{}^{2}+c_{12}c_{11}\right)}^{2}}\right)}^{3}\left(-2c_{22}^{2}+\omega_{2}{}^{2}c_{32}\right)}{8\omega_{2}{}^{10}c_{11}^{3}},\\ b_{1}=\frac{-\omega_{1}{}^{2}+\omega_{2}{}^{2}+c_{12}c_{11}+\sqrt{4\omega_{2}{}^{2}c_{11}c_{12}+{\left(\omega_{1}{}^{2}-\omega_{2}{}^{2}+c_{12}c_{11}\right)}^{2}}}{2c_{11}},\\ b_{3}=\frac{\left(-\omega_{1}{}^{2}+\omega_{2}{}^{2}-c_{12}c_{11}+\sqrt{4c_{12}\omega_{2}{}^{2}c_{11}+{\left(\omega_{1}{}^{2}-\omega_{2}{}^{2}+c_{12}c_{11}\right)}^{2}}\right)c_{21}}{c_{11}\left(\omega_{1}{}^{2}+\omega_{2}{}^{2}+c_{12}c_{11}-\sqrt{4c_{12}\omega_{2}{}^{2}c_{11}+{\left(\omega_{1}{}^{2}-\omega_{2}{}^{2}+c_{12}c_{11}\right)}^{2}}\right)},\\ b_{5}=\frac{\left(-\omega_{1}{}^{2}+\omega_{2}{}^{2}-c_{12}c_{11}+\sqrt{4c_{12}\omega_{2}{}^{2}c_{11}+{\left(\omega_{1}{}^{2}-\omega_{2}{}^{2}+c_{12}c_{11}\right)}^{2}}\right)c_{31}}{c_{11}\left(\omega_{1}{}^{2}+\omega_{2}{}^{2}+c_{12}c_{11}-\sqrt{4c_{12}\omega_{2}{}^{2}c_{11}+{\left(\omega_{1}{}^{2}-\omega_{2}{}^{2}+c_{12}c_{11}\right)}^{2}}\right)} \end{array}$$
mode 2:(57)$$\begin{array}{l} a_{1}=a_{3}=a_{5}=b_{2}=b_{4}=b_{6}=0,\\ a_{2}=\frac{\omega_{1}{}^{2}-\omega_{2}{}^{2}+c_{11}c_{12}+\sqrt{4\omega_{2}{}^{2}c_{11}c_{12}+{\left(\omega_{1}{}^{2}-\omega_{2}{}^{2}+c_{11}c_{12}\right)}^{2}}}{2\omega_{2}{}^{2}c_{11}},\\ a_{6}=-\frac{{\left(\omega_{1}{}^{2}-\omega_{2}{}^{2}+c_{11}c_{12}+\sqrt{4\omega_{2}{}^{2}c_{11}c_{12}+{\left(\omega_{1}{}^{2}-\omega_{2}{}^{2}+c_{11}c_{12}\right)}^{2}}\right)}^{3}\left(-2c_{22}^{2}+\omega_{2}{}^{2}c_{32}\right)}{8\omega_{2}{}^{10}c_{11}^{3}}\\ a_{4}=-\frac{{\left(\omega_{1}{}^{2}-\omega_{2}{}^{2}+c_{11}c_{12}+\sqrt{4\omega_{2}{}^{2}c_{11}c_{12}+{\left(\omega_{1}{}^{2}-\omega_{2}{}^{2}+c_{11}c_{12}\right)}^{2}}\right)}^{2}c_{22}}{4\omega_{2}{}^{6}c_{11}^{2}},\\ b_{1}=\frac{-\omega_{1}{}^{2}+\omega_{2}{}^{2}+c_{11}c_{12}-\sqrt{4\omega_{2}{}^{2}c_{11}c_{12}+{\left(\omega_{1}{}^{2}-\omega_{2}{}^{2}+c_{11}c_{12}\right)}^{2}}}{2c_{11}}\\ b_{3}=-\frac{\left(\omega_{1}{}^{2}-\omega_{2}{}^{2}+c_{11}c_{12}+\sqrt{4\omega_{2}{}^{2}c_{11}c_{12}+{\left(\omega_{1}{}^{2}-\omega_{2}{}^{2}+c_{11}c_{12}\right)}^{2}}\right)c_{21}}{c_{11}\left(\omega_{1}{}^{2}+\omega_{2}{}^{2}+c_{11}c_{12}+\sqrt{4\omega_{2}{}^{2}c_{11}c_{12}+{\left(\omega_{1}{}^{2}-\omega_{2}{}^{2}+c_{11}c_{12}\right)}^{2}}\right)},\\ b_{5}=-\frac{\left(\omega_{1}{}^{2}-\omega_{2}{}^{2}+c_{11}c_{12}+\sqrt{4\omega_{2}{}^{2}c_{11}c_{12}+{\left(\omega_{1}{}^{2}-\omega_{2}{}^{2}+c_{11}c_{12}\right)}^{2}}\right)c_{31}}{c_{11}\left(\omega_{1}{}^{2}+\omega_{2}{}^{2}+c_{11}c_{12}+\sqrt{4\omega_{2}{}^{2}c_{11}c_{12}+{\left(\omega_{1}{}^{2}-\omega_{2}{}^{2}+c_{11}c_{12}\right)}^{2}}\right)} \end{array}$$

Hence, in the modal motion, the following relation is satisfied:(58)$$\begin{array}{l} x_{1}=u,\;y_{1}=v,\\ x_{2}=a_{2}v+a_{4}v^{2}+a_{6}v^{3},\;y_{2}=b_{1}u+b_{3}u^{3}+b_{5}u^{3} \end{array}$$

Considering Equations (56) and (57), the decoupled nonlinear equations can be obtained by substituting Equation (58) into Equation (46) as follows:(59)$$\ddot{u}+\omega_{n}{}^{2}u+e_{2}u^{2}+e_{3}u^{3}=0$$
where
(60)$$\omega_{n}{}^{2}=\omega_{1}{}^{2}+c_{11}b_{1},\;e_{2}=c_{11}b_{3}+c_{21},\;e_{3}=c_{11}b_{5}+c_{31}$$

According to the method of multi-scale, we introduce the new time scales:(61)$$T_{n}=\varepsilon^{n}t\;n=0,1,2\ldots$$
where *ε* is introduced as a small nondimensional parameter.

The derivative of time can be expressed in the form of the following operator by the partial derivative of *T_n_*:(62)$$\left\{ \begin{array}{l} \frac{d}{dt}=\frac{\partial }{\partial T_{0}}\frac{dT_{0}}{dt}+\frac{\partial }{\partial T_{1}}\frac{dT_{1}}{dt}+\frac{\partial }{\partial T_{2}}\frac{dT_{2}}{dt}\ldots =D_{0}+\varepsilon D_{1}+\varepsilon^{2}D_{2}\ldots \\ \frac{d^{2}}{dt^{2}}=D_{0}{}^{2}+2\varepsilon D_{0}D_{1}+\varepsilon^{2}(D_{0}{}^{2}+2D_{0}D_{2})+\ldots \end{array}\right.$$

Adopting the second-order approximation, the solution of Equation (59) is expressed as:(63)$$u(t)=u_{0}(T_{0},T_{1},T_{2})+\varepsilon u_{1}(T_{0},T_{1},T_{2})+\varepsilon^{2}u_{2}(T_{0},T_{1},T_{2})$$

Substituting Equations (62) and (63) into Equation (59) and comparing the same power coefficients of *ε*, yield:

$O(\varepsilon^{0})$:(64)$$D_{0}{}^{2}u_{0}+\omega_{n}{}^{2}u_{0}=0$$$O(\varepsilon^{1})$:(65)$$\omega_{n}{}^{2}u_{1}+D_{0}{}^{2}u_{1}=-e_{2}u_{0}{}^{2}-2D_{0}D_{1}u_{0}$$$O(\varepsilon^{2})$:(66)$$\omega_{n}{}^{2}u_{2}+D_{0}{}^{2}u_{2}=-e_{3}u_{0}{}^{3}-2e_{2}u_{0}u_{1}-D_{1}{}^{2}u_{0}-2D_{0}D_{2}u_{0}-2D_{0}D_{1}u_{1}$$

The general solution of Equation (64) can be expressed as:(67)$$u_{0}(T_{0},T_{1},T_{2})=A_{11}(T_{1},T_{2})\exp\left(i\omega_{n}T_{0}\right)+cc$$
where *A*_11_ is the amplitude.

Substituting Equation (67) into Equation (65), we have:(68)$$e_{2}e^{2i\omega_{n}T_{0}}A_{11}{\left(T_{1},T_{2}\right)}^{2}+2e_{2}A_{11}\left(T_{1},T_{2}\right)\bar{A_{11}}\left(T_{1},T_{2}\right)+2\omega_{n}{\mathrm{ie}}^{i\omega_{n}T_{0}}D_{1}A_{11}\left(T_{1},T_{2}\right)+cc=0$$
where $\exp\left(\pm i\omega_{n}T_{0}\right)$ is the secular term and the condition for eliminating the secular term can be obtained as follows:(69)$$2i\omega_{n}D_{1}A_{11}\left(T_{1},T_{2}\right)=0$$
which indicates that *A*_11_ is just a function of *T*_2_.

The solution of Equation (68) can be expressed as:(70)$$u_{1}\left(T_{0},T_{2}\right)=-\frac{e^{2i\omega_{n}T_{0}}e_{2}A_{11}{\left(T_{2}\right)}^{2}}{\omega_{n}{}^{2}}-\frac{2e_{2}A_{11}\left(T_{2}\right)\bar{A_{11}}\left(T_{2}\right)}{\omega_{n}{}^{2}}-\frac{e^{-2i\omega_{n}T_{0}}e_{2}\bar{A_{11}}{\left(T_{2}\right)}^{2}}{\omega_{n}{}^{2}}$$

Substituting Equations (67) and (69) into Equation (66), we obtain:(71)$$\begin{array}{l} \omega_{n}{}^{2}u_{2}+D_{0}{}^{2}u_{2}=-e^{3i\omega_{n}T_{0}}\left(e_{3}A_{11}{\left(T_{2}\right)}^{3}-\frac{2e_{2}{}^{2}A_{11}{\left(T_{2}\right)}^{3}}{\omega_{n}{}^{2}}\right)-\\ e^{i\omega_{n}T_{0}}\left(3e_{3}A_{11}{\left(T_{2}\right)}^{2}\bar{A_{11}}\left(T_{2}\right)-\frac{6e_{2}{}^{2}A_{11}{\left(T_{2}\right)}^{2}\bar{A_{11}}\left(T_{2}\right)}{\omega_{n}{}^{2}}+2i\omega_{n}D_{2}A_{11}\left(T_{2}\right)\right)+cc \end{array}$$

To eliminate the secular term of Equation (71), we let:(72)$$3e_{3}A_{11}{\left(T_{2}\right)}^{2}\bar{A_{11}}\left(T_{2}\right)-\frac{6e_{2}{}^{2}A_{11}{\left(T_{2}\right)}^{2}\bar{A_{11}}\left(T_{2}\right)}{\omega_{n}{}^{2}}+2i\omega_{n}D_{2}A_{11}\left(T_{2}\right)=0$$

Express *A*_11_ as a polar coordinate in the form:(73)$$A_{11}\left(T_{2}\right)=\frac{1}{2}a_{1}\left(T_{2}\right)\exp\left(i\theta_{1}\left(T_{2}\right)\right),\;\bar{A_{11}}\left(T_{2}\right)=\frac{1}{2}a_{1}\left(T_{2}\right)\exp\left(-i\theta_{1}\left(T_{2}\right)\right)$$
where *a*_1_ and *θ*_1_ denote the real amplitude and the initial phase, respectively.

Substituting Equation (73) into Equation (72) and separating the real and imaginary parts, we have:(74)$$a_{1}{}^{\prime }\left(T_{2}\right)=0,\theta_{1}{}^{\prime }\left(T_{2}\right)=\frac{3\left(-2e_{2}{}^{2}+e_{3}\omega_{n}{}^{2}\right)a_{1}{\left(T_{2}\right)}^{2}}{8\omega_{n}{}^{3}}\to a_{1}=\Gamma ,\theta_{1}=\frac{3\left(-2e_{2}{}^{2}+e_{3}\omega_{n}{}^{2}\right)a_{1}{}^{2}}{8\omega_{n}{}^{3}}T_{2}+C_{2}$$
where Γ and *C*_2_ are constants.

When the phase difference *C*_2_ = 0, then the approximate solution of Equation (63) can be written as:(75)$$\begin{array}{l} u_{0}=A_{11}(T_{2})\exp\left(i\omega_{n}T_{0}\right)+cc\\ \;=\frac{1}{2}\Gamma \exp[-i\omega_{n}T_{0}-\frac{3i\Gamma^{2}\left(-2e_{2}{}^{2}+e_{3}\omega_{n}{}^{2}\right)\varepsilon^{2}}{8\omega_{n}{}^{3}}T_{0}]+\frac{1}{2}\Gamma \exp[i\omega_{n}T_{0}+\frac{3i\Gamma^{2}\left(-2e_{2}{}^{2}+e_{3}\omega_{n}{}^{2}\right)\varepsilon^{2}}{8\omega_{n}{}^{3}}T_{0}]\\ \;=\Gamma \cos\left[-\left(\frac{3\Gamma^{2}\left(-2e_{2}{}^{2}+e_{3}\omega_{n}{}^{2}\right)\varepsilon^{2}}{8\omega_{n}{}^{3}}+\omega_{n}\right)T_{0}\right] \end{array}$$

The nonlinear natural frequencies can also be obtained as follows:(76)$$\omega_{n1,n2}=\omega_{n}+\frac{3\Gamma^{2}\left(-2e_{2}{}^{2}+e_{3}\omega_{n}{}^{2}\right)\varepsilon^{2}}{8\omega_{n}{}^{3}}$$

Substituting Equations (56), (57), and (60) into Equation (76) yields two nonlinear natural frequencies without damping and external excitation:(77)$$\begin{array}{l} \omega_{n1}=\frac{\sqrt{2}\omega_{2}{}^{4}\left(8\omega_{1}{}^{4}+3\Gamma^{2}\varepsilon^{2}\left(-2c_{21}^{2}+\omega_{1}{}^{2}c_{31}\right)\right)}{{\left(\omega_{1}{}^{2}+\omega_{2}{}^{2}+c_{11}c_{12}-\sqrt{\left({\left(\omega_{1}-\omega_{2}\right)}^{2}+c_{11}c_{12}\right)\left({\left(\omega_{1}+\omega_{2}\right)}^{2}+c_{11}c_{12}\right)}\right)}^{3/2}}\\ \times \frac{1}{{\left(\omega_{1}{}^{2}+\omega_{2}{}^{2}+c_{11}c_{12}+\sqrt{\left({\left(\omega_{1}-\omega_{2}\right)}^{2}+c_{11}c_{12}\right)\left({\left(\omega_{1}+\omega_{2}\right)}^{2}+c_{11}c_{12}\right)}\right)}^{2}}\\ \omega_{n2}=\frac{\sqrt{2}\omega_{2}{}^{4}\left(8\omega_{1}{}^{4}+3\Gamma^{2}\varepsilon^{2}\left(-2c_{21}^{2}+\omega_{1}{}^{2}c_{31}\right)\right)}{{\left(\omega_{1}{}^{2}+\omega_{2}{}^{2}+c_{11}c_{12}-\sqrt{\left({\left(\omega_{1}-\omega_{2}\right)}^{2}+c_{11}c_{12}\right)\left({\left(\omega_{1}+\omega_{2}\right)}^{2}+c_{11}c_{12}\right)}\right)}^{2}}\;\\ \times \frac{1}{{\left(\omega_{1}{}^{2}+\omega_{2}{}^{2}+c_{11}c_{12}+\sqrt{\left({\left(\omega_{1}-\omega_{2}\right)}^{2}+c_{11}c_{12}\right)\left({\left(\omega_{1}+\omega_{2}\right)}^{2}+c_{11}c_{12}\right)}\right)}^{3/2}} \end{array}$$

To verify the correctness of the IMM, the results of IMM and DQM are compared, as shown in Figure 12. Note that the influence of electrostatic nonlinearity is not considered at this time. Here, *V_DC_*_1_ = *V_DC_*_2_ = 0.42 V, *α*_4_ = *α*_5_ = 0.5, 0. It can be seen that the results of the two methods are consistent.

Figure 13 shows the variation of two nonlinear frequencies with motion amplitude Γ for different angular velocities. The system parameters are set as: *α*_4_ = *α*_5_ = −0.3, 0.3, *V_DC_*_1_ = *V_DC_*_2_ = 1.0 V, Ω = 1 rad/s, 1000 rad/s, 2000 rad/s. The dashed line represents the linear frequency, and the solid line represents the nonlinear frequency. It can be found that, due to the softening characteristics of electrostatic force, the nonlinear frequency decreases and gradually moves away from the linear frequency with the increase in the motion amplitude. In addition, as the angular velocity increases, *ω_n_*_1_ decreases, and *ω_n_*_2_ increases gradually. A comparison between *α*_4_ = *α*_5_ = −0.3 and *α*_4_ = *α*_5_ = 0.3 reveals that the nonlinear frequency shift of the micro-gyroscope with an increasing motion amplitude is smaller for *α*_4_ = *α*_5_ = −0.3. That is, for the same motion amplitude Γ, the micro-gyroscope is less affected by nonlinearity at *α*_4_ = *α*_5_ = −0.3. This is due to the fact that the shape factor decreases, and the tapered beam becomes progressively stiffer, which can resist the softening characteristics of electrostatic force nonlinearity.

The variation curves of the linear and nonlinear frequency with angular velocity for different shape factors are given in Figure 14. The system parameters are set as: *α*_4_ = *α*_5_ = −0.3, 0, 0.3, *V_DC_*_1_ = *V_DC_*_2_ = 1.0 V, Γ = 1. It can be found that the nonlinear frequency of the micro-gyroscope with a shape factor of 0.3 has the largest shift with respect to the linear frequency. Thus, the micro-gyroscope with *α*_4_ = *α*_5_= 0.3 has the strongest nonlinearity for the same motion amplitude. The sensitivity based on nonlinear frequency detection is less than the linear case at this time, as shown in Figure 14d. As the shape factors decrease gradually, the nonlinear frequency shift decreases and the nonlinearity gradually weakens, as shown in Figure 14b,c. When *α*_4_ = *α*_5_ = 0, the sensitivity of the micro-gyroscope is less affected by the nonlinearity. When *α*_4_ = *α*_5_ = −0.3, the nonlinear frequency shift of the micro-gyroscope is the smallest, and the sensitivity performance is not affected by nonlinearity at this time. It indicates that the influence of nonlinearity on the sensitivity of the FM micro-gyroscope can be attenuated by using a negative shape factor.

Figure 15 illustrates the effects of different bias DC voltages on the sensitivity of the micro-gyroscope with shape factors of 0.3, 0, and −0.3, respectively. Here, the motion amplitude Γ = 1. In Figure 15a, it can be found that the sensitivity performance of the micro-gyroscope gradually decreases as the DC voltage gradually increases. This is due to the fact that the DC voltage is proportional to the electrostatic force nonlinearity. The higher the DC voltage, the stronger the electrostatic force nonlinearity. The nonlinearity can reduce the sensitivity performance of the FM micro-gyroscope. For the micro-gyroscope with *α*_4_ = *α*_5_ = 0.3, the system sensitivity is already extremely small at *V_DC_*_1_ = *V_DC_*_2_ = 2 V, and it is not advisable to increase the voltage further. For micro-gyroscopes with *α*_4_ = *α*_5_ = 0 and *α*_4_ = *α*_5_ = −0.3, the influence of nonlinearity on the sensitivity decreases as the shape factor decreases. At the same bias DC voltage, the micro-gyroscope with the negative shape factor can better restrain the influence of nonlinearity. Therefore, in order to ensure the sensitivity performance and linearity of the FM micro-gyroscope, a smaller bias DC voltage should be selected. Meanwhile, the tapered beam structure with a negative shape factor can not only reduce the effect of nonlinearity, but can also obtain a larger dynamic range.

## 7. Conclusions

In this paper, the influence of electrostatic force nonlinearity on the sensitivity performance of a class of tapered beam FM micro-gyroscopes is investigated. The pull-in voltage and natural frequency of the micro-gyroscope are analyzed using DQM and the effect of electrostatic force nonlinearity on the FM micro-gyroscope is investigated by using IMM. Recommendations are given for guidance on the structural design of the micro-gyroscope and the principle of parameter selection, which improves the micro-gyroscope performance overall. It promotes the possibility of micro-gyroscope applications in medical instruments, commercial attitude heading reference systems, guided munitions, and other high-precision fields. The main conclusions are as follows:
(1)The shape factor can be used to adjust the dynamic detection range of the FM micro-gyroscope without reducing the system sensitivity.(2)Unequal bias DC voltages in the drive and sense directions, and unequal width and thickness shape factors, can break the symmetry of the micro-gyroscope. This leads to a frequency mismatch of the micro-gyroscope and the reduced sensitivity of the system. The effect caused by the asymmetry should be reduced.(3)As the DC voltage in the drive and sense directions increases, the electrostatic force nonlinearity gradually increases, and the sensitivity performance decreases significantly. For the same DC voltage and motion amplitude, the sensitivity performance of the micro-gyroscope with a negative shape factor is least affected by the nonlinearity. This is because the smaller the shape factor is, the greater the structural stiffness of the tapered beam can effectively reduce the softening characteristics of the electrostatic force nonlinearity.

## Figures and Tables

**Figure 1 micromachines-14-00211-f001:**
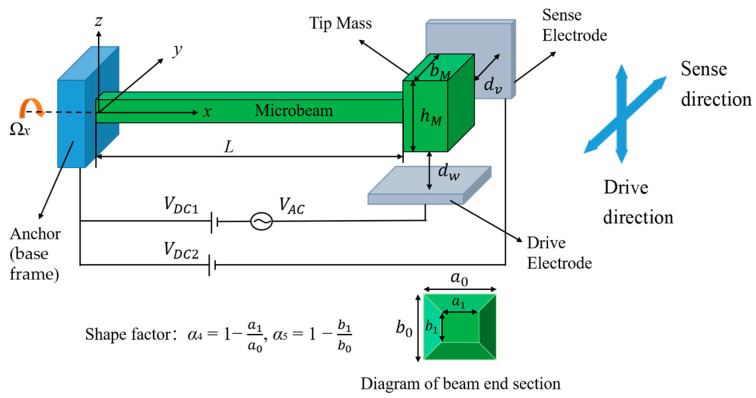
Schematic diagram of the tapered beam micro-gyroscope.

**Figure 2 micromachines-14-00211-f002:**
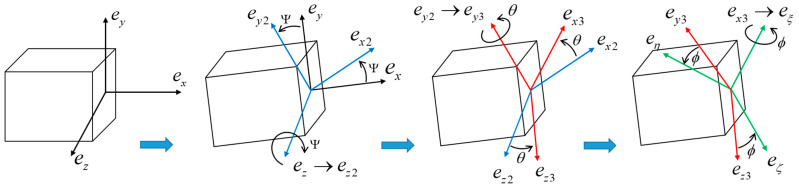
Schematic diagram of Eulerian angle transformation.

**Figure 3 micromachines-14-00211-f003:**
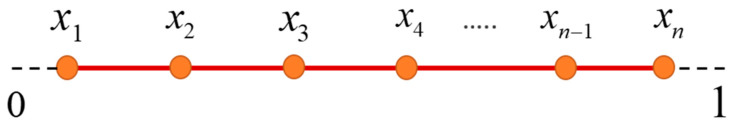
Schematic diagram of nodes distribution on the interval [0, 1].

**Figure 4 micromachines-14-00211-f004:**
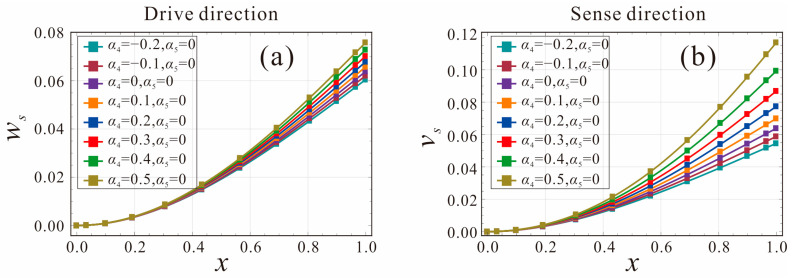
Effect of shape factors *α*_4_ and *α*_5_ on the static deflection of the micro-gyroscope: (**a**) drive direction; (**b**) sense direction.

**Figure 5 micromachines-14-00211-f005:**
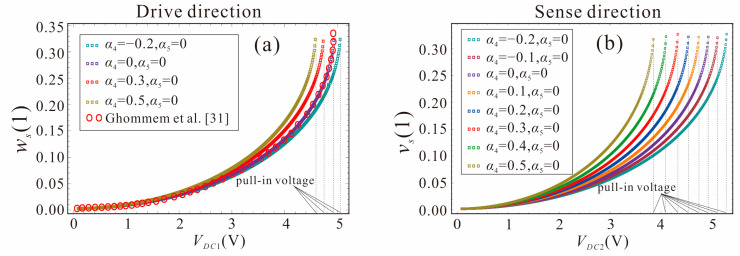
Variation curve of maximum displacement at the end of micro-gyroscope with DC voltage: (**a**) Drive direction, (**b**) Sense direction.

**Figure 6 micromachines-14-00211-f006:**
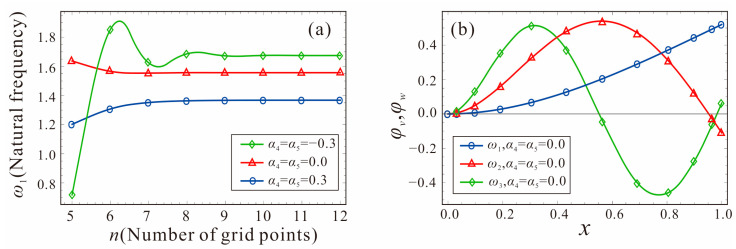
Natural frequency and the first three orders of mode shapes of the tapered beam: (**a**) Natural frequency, (**b**) Mode shapes.

**Figure 7 micromachines-14-00211-f007:**
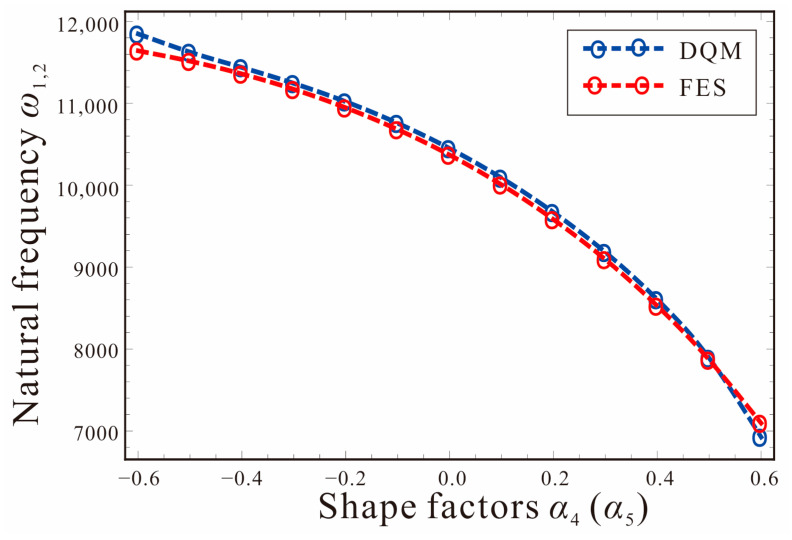
Comparison of the first two orders natural frequencies of the tapered beam between DQM and FES.

**Figure 8 micromachines-14-00211-f008:**
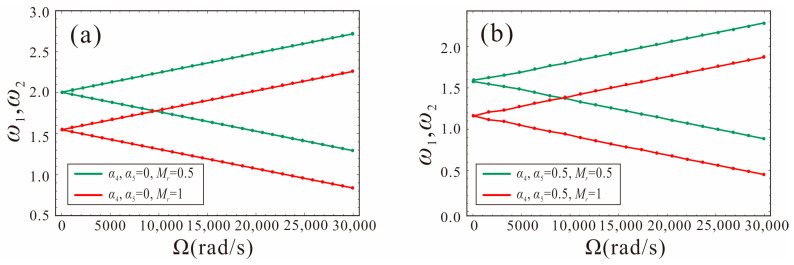
Variation of the natural frequency of the tapered beam micro-gyroscope with the angular velocity: (**a**) *α*_4_ = *α*_5_ = 0, (**b**) *α*_4_ = *α*_5_ = 0.5.

**Figure 9 micromachines-14-00211-f009:**
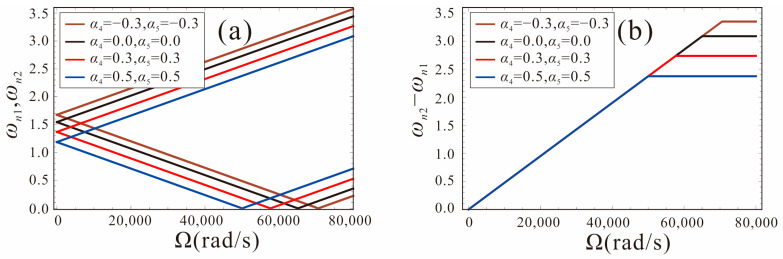
The first order natural frequency versus input angular velocity of the system with different shape factors.

**Figure 10 micromachines-14-00211-f010:**
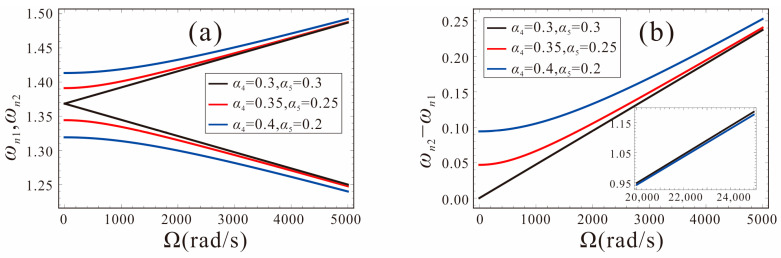
Effects of angular velocity on the natural frequency for the case of different shape factors.

**Figure 11 micromachines-14-00211-f011:**
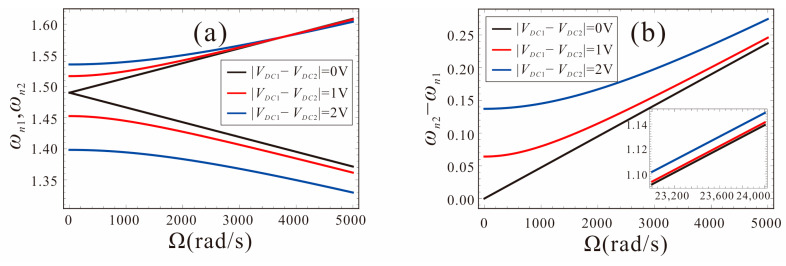
Effects of bias DC voltage asymmetry in the drive and sense directions on the sensitivity.

**Figure 12 micromachines-14-00211-f012:**
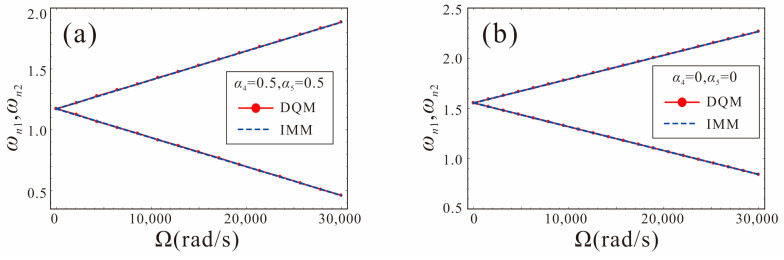
Comparison between numerical solution of DQM and analytical solution of IMM: (**a**) shape factors are *α*_4_ = *α*_5_ = 0.5; (**b**) shape factors are *α*_4_ = *α*_5_ = 0.

**Figure 13 micromachines-14-00211-f013:**
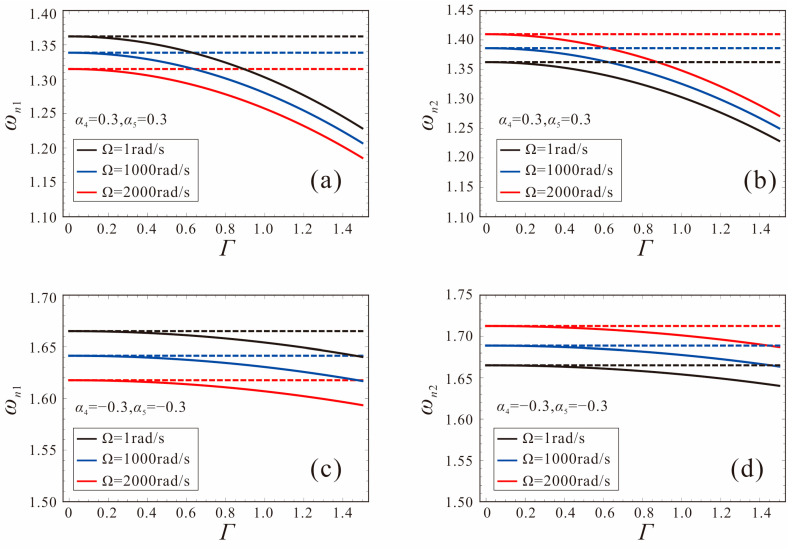
Variation curves of the nonlinear frequency with motion amplitude. (**a**,**b**): shape factors are *α*_4_ = *α*_5_ = 0.3; (**c**,**d**): shape factors are *α*_4_ = *α*_5_ = −0.3.

**Figure 14 micromachines-14-00211-f014:**
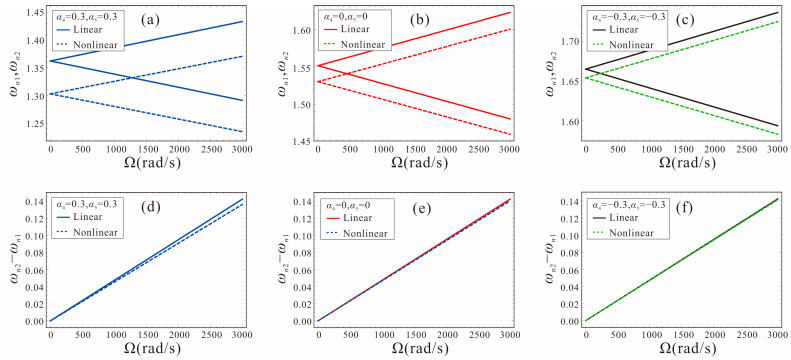
Linear frequency and nonlinear frequency variation curves of micro-gyroscope for different shape factors. (**a,d**): shape factors are *α*_4_ = *α*_5_ = 0.3; (**b,e**): shape factors are *α*_4_ = *α*_5_ = 0; (**c,f**): shape factors are *α*_4_ = *α*_5_ = −0.3.

**Figure 15 micromachines-14-00211-f015:**
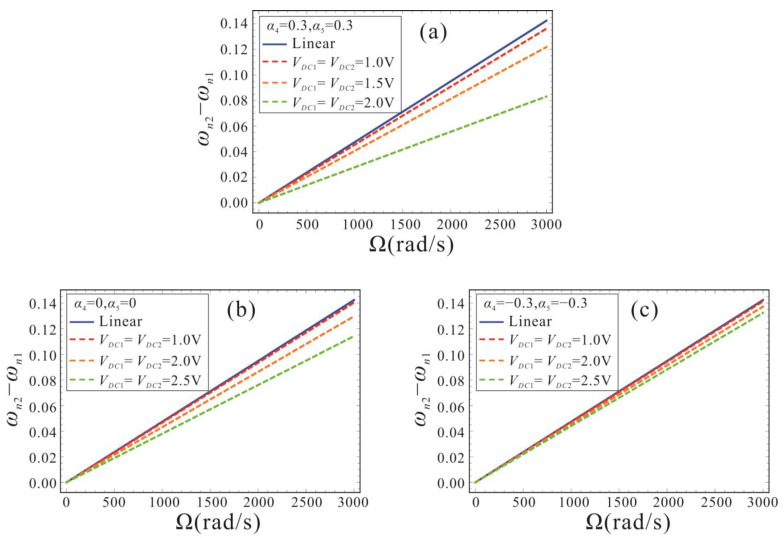
Sensitivity performance of micro-gyroscope with different bias DC voltage. (**a**): shape factors are *α*_4_ = *α*_5_ = 0.3; (**b**): shape factors are *α*_4_ = *α*_5_ = 0; (**c**): shape factors are *α*_4_ = *α*_5_ = −0.3.

**Table 1 micromachines-14-00211-t001:** The natural frequencies of the ROM of micro-gyroscope.

				*n* _2_					
*α* _4_	*α* _5_	Direction	*ω*	6	8	10	12	14	16
0	0	Drive	10,448	10,532	10,448	10,448	-	-	-
0	0.5	Drive Sense	8587 10,137	11,905 10,739	9178 10,185	8658 10,141	8609 10,138	8600 10,136	8589 -
0.5	0.5	Drive	7963	15,755	10,076	8296	8070	8020	7964

**Table 2 micromachines-14-00211-t002:** Angular gain factors for FM micro-gyroscope with different shape factors.

*α*_4_, *α*_5_	*α*_4_ = 0.5, *α*_5_ = 0.5	*α*_4_ = 0.3, *α*_5_ = 0.3	*α*_4_ = 0.4, *α*_5_ = 0.2	*α*_4_ = 0.5, *α*_5_ = 0.2	*α*_4_ = 0.5, *α*_5_ = 0.1	*α*_4_ = 0.5, *α*_5_ = 0
*κ*	0.99999	0.99999	0.99427	0.98555	0.97667	0.96686

## Data Availability

The data presented in this study are available on request from the corresponding author.

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
