# Peer review of "Influence of Electrostatic Force Nonlinearity on the Sensitivity Performance of a Tapered Beam Micro-Gyroscope Based on Frequency Modulation"

_micromachines, 2023, doi:10.3390/mi14010211_

Round 1

Reviewer 1 Report

In this paper, the impact of nonlinearity in electrostatic force on the performance of a tapered beam micro-gyroscopes is investigated. Hamilton principle was used to derive the equation of motion. The results show that asymmetry between the width and thickness and unequal DC voltages in both the drive and sense directions decrease the FM performance of the micro-gyroscope. The topic is interesting. However, there are several concerns about this manuscript which needs to be addressed before its acceptance.

1)     Abstract sounds unclear. Rewriting/rephrasing of the abstract is recommended.   

2)     It is recommended to avoid compounding of references.

3)     Introduction section lack critical review of the previous literature.

4)     It is recommended to briefly explain some of the terminologies (e.g. amplitude modulation, frequency modulation etc) before in discussing them.   

5)     Motivation of the current study in terms of the shortcoming in the literature is missing.

6)     Organization of the manuscript at the end of introduction section is recommended.

7)     It is recommended to replace the schematic diagram (Figure 1) with a better-quality figure.

8)     Description of the tapered beam given in the early part of section 2 needs a serious revision.

9)     Caption of Figure 4 is confusing. It is recommended to replace with more concise one.

10) What was the critical value of DC voltage?

11) At what point pull-in instability is expected?

12) The assumptions used in the model should be stated in the paper more clearly and rigorously.

13) An explanation of the boundary conditions for the modeling procedure is needed.

14) The key challenges should be identified and recommendations for work should be provided in the conclusion.

15) The linguistic quality of the paper is also weak.

Reviewer 2 Report

This paper studies a complex problem that gives much headache in real life. It is hard to reconcile all the variable parameters envolved in this micro-system, the critical sensitivity, the nonlinearity problems of the force while increasing voltage, the physical shape factors of the gyroscope, by accordingly modulating the frequency. Of course compromises appear.

The problem is clearly described, the mathematical approach correctly adapted to the concrete situation in cause.  The study properly considers the documentation in the references, approaching this problem in a very specific way. The physical phenomena is accordingly studied, denoting the well understanding of the problem.

Figures 4 and 5, allthough they are very clear from the mathematical point of view, they don’t seem to me intuitive at all, regarding the results of the measurements and tests. 

The other figures and computing results, seem to me clear and enough explained.

Regarding the final research, it seems though that the conclusions are pretty vague. So, after a thorough study, well done and explained, I’d expect more of a practical result or at least some concrete practical future applications of this study.

Round 2

Reviewer 1 Report

The manuscript has been significantly improved. All of my comments are well taken care of in the revised manuscript. Therefore, I am satisfied with the revision made by the authors, and would like to recommend it for publication in its current form.

Reviewer 3 Report

The manuscript has been significantly improved. And all previous comments have been carefully revised point by point. So I recommend to accept in present form.